



# Vertical transport of sediment-associated metals and cyanobacteria by ebullition in a stratified lake

Kyle Delwiche[1#], Junyao Gu[2], Harold Hemond[1] and Sarah P. Preheim[2]

[1]Department of Civil and Environmental Engineering, Massachusetts Institute of Technology, Cambridge, MA, U.S.A

[2]Department of Environmental Health and Engineering, Johns Hopkins University, Baltimore, MD, U.S.A

[#]Current affiliation: Harvard John A. Paulson School of Engineering and Applied Sciences, Harvard University, Cambridge, MA, U.S.A

*Correspondence to*: Sarah Preheim (sprehei1@jhu.edu) and Kyle Delwiche (kdelwiche@seas.harvard.edu)

**Abstract.** Bubbles adsorb and transport particulate matter both in industrial and marine systems. While methane-containing
bubbles emitted from anoxic sediments are found extensively in aquatic ecosystems, relatively little attention has been paid to the possibility that such bubbles transport particle-associated chemical or biological material from sediments to surface waters of freshwater lakes. We quantified transport of particulate material from sediments to the surface by bubbles in Upper Mystic Lake, MA and in a 15 m tall experimental column. Vertical particle transport was positively correlated with the volume of gas bubbles released from the sediment. Particles transported by bubbles originated almost entirely in the
sediment, rather than being scavenged from the water column. Concentrations of arsenic, chromium, lead, and cyanobacterial cells in bubble-transported particulate material were similar to those of bulk sediment, and particles were transported from depths exceeding 15 m, resulting in daily fluxes as large as 0.18 µg of arsenic m$^{-2}$ and 2 x 10$^4$ cyanobacterial cells m$^{-2}$ in the strongly stratified Upper Mystic Lake. While bubble-facilitated arsenic transport currently appears to be a modest component of total arsenic cycling in this lake, bubble-facilitated cyanobacterial transport could
comprise as much as 17% of recruitment in this lake and may thus be of particular importance in large, deep, stratified lakes.
## 1 Introduction

Deterioration of water quality is wide-spread and are expected to become more acute with increased urbanization and climate-change. In a 2012 national assessment, 15.2% of surveyed lakes in the U.S. were found to have problematic cyanobacteria cell numbers, an 8% increase over the 2007 assessment (U.S. Environmental Protection Agency, 2016).

Concentrations of $10^5$ cyanobacteria cells $mL^{-1}$ are considered to present a risk of both acute and chronic health effects (Backer, 2002), and many states, including Massachusetts, issue public health warnings when cyanobacteria reach cell concentrations approaching these values for recreational water bodies. Metals are also important contaminants in freshwater systems because of their persistence and toxicity (Bronmark and Hansson, 2002). In 2004, 1.5 million lake-acres in the U.S. were impaired by metals, such as lead, chromium and arsenic (Environmental Protection Agency, 2004). An improved

understanding of the sources and mechanisms of transport of these substances within lake ecosystems can help predict the fate of contaminants and aid remediation efforts.

Because sediments are typically major repositories of contaminants (Nriagu et al., 1996;Pan and Wang, 2012;Taylor and Owens, 2009), processes leading to mobilization of these constituents from sediments are important to understand. Metals can be mobilized from sediments via solubilization by oxidation-reduction reactions, and by sediment resuspension

or bioturbation (Calmano et al., 1993;Eggleton and Thomas, 2004;Schaller, 2014). However, transport to surface waters of contaminants mobilized from the sediment is affected by lake hydrodynamic conditions, notably stratification (Hicks et al., 1994). Likewise, over-wintering cyanobacteria and algae concentrated in the sediments are recruited to the surface through germination, wind-induced resuspension, or bioturbation (Ramm et al., 2017;Verspagen et al., 2004;Stahl-Delbanco and Hansson, 2002). In some cases, the number of resting cells in sediment can be predictive of the severity of subsequent bloom

events  (Anderson et al., 2005).  Verspagen et. al. (2005) showed that recruitment from sediments of the potentially toxic cyanobacterium *Microcystis* was a major driver of the summer bloom (Verspagen et al., 2005). Cyanobacterial recruitment to surface waters from deep sediments is expected to be inhibited by stratification, low oxygen concentration, and low light levels (Ramm et al., 2017).

An alternative mechanism for vertical transport of metals and cells from sediment to surface water could be bubble-

facilitated transport.  Bubbling from anoxic sediments, driven by methanogenesis, is widespread in freshwater systems, and

bubbles are known to be effective particle transporters.  Bubble particle flotation, a process by which amphiphilic particles

attach to a bubble's gas-water interface and are transported upwards during bubble rise, is used extensively in industry for

applications such as separating valuable minerals from gangue (Min et al., 2008;Rodrigues and Rubio, 2007), removing ink

during paper recycling (Vashisth et al., 2011), recovering desirable proteins and microorganisms from industrial bioreactors

(Schugerl, 2000), and treating wastewaters (Aldrich and Feng, 2000;Lin and Lo, 1996;Rubio et al., 2002). Bubble-mediated

particle transport also occurs in the open ocean and contributes to an accumulation of surface-active particles at the surface

of the ocean (Aller et al., 2005;Blanchard, 1975;Wallace et al., 1972), as bubbles are injected into the ocean surface by

breaking waves, and their particle burden is obtained by scavenging of particles as the bubbles rise (Liss, 1975).

Despite this previous work, little is known about the importance of particle transport by bubbles in freshwater

systems.  Bubbles produced by methanogenesis in anoxic sediments are prevalent in freshwater systems, and bubbles are

released to the surface during drops in hydrostatic pressure, sediment disturbance, or upon sufficient gas accumulation

(Chanton et al., 1989;Joyce and Jewell, 2003;Scandella et al., 2011;Liu et al., 2016;Maeck et al., 2014;Varadharajan and

Hemond, 2012). Bubble flotation could thus potentially provide a chemical and biological link from deep water to surface

waters that would otherwise not occur through advective or eddy-diffusive transport alone.  Additionally, the relatively rapid

rise time of bubbles limits the time available for oxidation reactions, and suggests that particulate matter from the

hypolimnion could reach the lake surface in a reduced state, with possible consequences for both toxicity and reactivity.

Some evidence does suggest that bubbles can transport polycyclic aromatic hydrocarbons (Viana et al., 2012) and

manufactured gas plant tar from sediments (McLinn and Stolzenburg, 2009). Additional work has shown that bubble-

mediated transport of microorganisms including methane oxidizing bacteria (MOB) is an important mechanism connecting

benthic and pelagic populations at 10 m water depth (Schmale et al., 2015). However, researchers in the previous study were

unable to quantify the importance of bubble-mediated transport to overall recruitment of pelagic MOB populations, and the

full extent of bubble particle flotation in aquatic systems remains unknown.

The present study is motivated by authors' observations of particle accumulations associated with bubbling events at Upper Mystic Lake (UML), where bursting bubbles often left black particles distributed on the water surface in a ring pattern (Fig. S1). Particles were also observed at the air-water interface in bubble traps during long-term deployments (data not shown). The significant volumes of gas observed to bubble from UML during previous studies (Delwiche and Hemond, 2017;Varadharajan and Hemond, 2012), together with strong thermal stratification suppressing other mechanisms of sediment transport to the surface, led to the hypothesis that bubbles could serve as a relatively important mode of particle transport from the sediment to the water surface. This potential transport pathway could be of greatest relative importance for metal and cyanobacteria transport in eutrophic, deep, stratified lakes, such as UML.

In the present study, we quantified particle transport by bubbles in UML, an urban lake with a history of sediment contamination. We also used a 15 m tall bubble column to study bubble-mediated particle transport under controlled lab conditions. Given the expected importance of both bubble size and total bubble volume, we used a bubble size sensor (Delwiche et al., 2015;Delwiche and Hemond, 2017) to measure bubble diameter distributions both in the lake and in the laboratory. We address the following questions:

1. How much sediment is transported to the surface through ebullition?

2. How does bubble-mediated sediment transport contribute to metal cycling?

3. How does bubble-mediated sediment transport contribute to cyanobacteria recruitment to the upper water column?

**2 Methods**

**2.1 Upper Mystic Lake field site history**

UML in Arlington, MA is an urban, dimictic kettle lake with an average depth of 15 m, a maximum depth of 24 m, and a surface area of 0.58 km$^2$. The lake is used extensively for recreational and scientific purposes, and previous studies have characterized several aspects of methane ebullition (Delwiche and Hemond, 2017;Scandella et al., 2016;Varadharajan and Hemond, 2012) and microbial community structure and function (Preheim et al., 2016;Arora-Williams et al., 2018).
Chemical manufacturing and leather tanning industries during the late 1800s and 1900s released toxic metals such as arsenic, chromium, and lead that flowed into the lake and were deposited in the lake sediments. Sediment cores reveal a distinct layered pattern with peak metal/metalloid concentrations traceable to years of peak manufacturing or subsequent earth-moving (Spliethoff and Hemond, 1996). Additionally, high nutrient loading promotes the growth of algal and cyanobacterial blooms. A public health advisory was issued for UML as recently as July 2017 for cyanobacteria cell concentrations

>70,000 cells mL$^{-1}$ (https://www.arlingtonma.gov/Home/Components/News/News/4965/16 accessed on 06/05/2019).

Years of field observations at UML have provided a thorough picture of the typical hydrological conditions in the lake. Significant volumes of gas are produced from the sediments, which escape to the surface via ebullition, resulting in an average release rate of 22 ml of bubble volume m$^{-2}$ d$^{-1}$ (Varadharajan, 2009). From June - Oct., the oxycline and thermocline are typically found between 6-12 m and 3-9 m, respectively (Varadharajan, 2009;Delwiche and Hemond, 2017). The Secchi

depth in the lake is typically 2-3 m during the same period (Varadharajan, 2009). Light was sufficient for germination down to 12 m in another lake with a similar average Secchi depth (Ramm et al., 2017), thus we assume light does not limit cyanobacteria germination down to a depth of at least 12 m when estimating the impact of bubbling on cyanobacteria recruitment.

**2.2 Field sampling**

Bubble-transported particles were collected from both the laboratory column and the lake in 350 mL plastic sampling cups affixed either to the top of a custom bubble size sensor [sensor described previously (Delwiche et al., 2015;Delwiche and Hemond, 2017)], or to the top of a collection funnel (the bubble sensor was used in 2017 sampling; the funnel alone was used for sampling in 2018). The plastic sampling cup lid contained a barbed bulkhead fitting connected via flexible plastic

tubing to an on-off valve and a quick-release adapter (Fig. S2). The sampling cup, valve, and adapter were connected to the custom bubble size sensor or collection funnel with flexible tubing. All bubbles rising through the bubble size sensor or collection funnel entered the flexible tubing and rose into the sample cup. The sample cup lid contained a secondary valve to release water upon bubble entry. All sample cups were soaked in 5-10% HNO$_3$ for 24 hours and rinsed and filled with
Milli-Q water prior to use. Gas and associated particles accumulated in the collection cup during sampling, and were then

transported back to the lab for analysis.

On 17 October 2017 we sampled for bubble-mediated sediment mass fluxes and associated particulate metal fluxes in an area of the lake previously found to have relatively high ebullition rates [approx. 42.432 latitude, -71.151 longitude, and 16 m deep; (Delwiche and Hemond, 2017)]. This previous work showed that sediment ebullition rates from this location remain high from July to November, yet the water column remains stratified, preventing mixing from of the sediment to the

surface. Previous work at this particular location within the lake indicated that natural bubble fluxes were around 45 mL m$^{-2}$ day$^{-1}$ with high spatial and temporal variability (Delwiche and Hemond, 2017). Given the need to collect samples as soon after bubbling as possible to minimize potential changes in cyanobacteria population, and the difficulty with predicting flux from natural bubble events, we chose to trigger ebullition manually by dropping an 20 cm x 20 cm x 20 cm cinderblock anchor into the sediment. This procedure enabled us to collect multiple samples during a single field trip, with minimal time

for samples to change after collection in the sampling cup.

After bubble triggering, the bubble size sensor was positioned above the bubble plume and 1 m below the water surface. Bubbles exiting the sensor, together with any particles adhered to the bubble/water interface, were collected in the sample cup described previously. Several anchor drops within an area of approximately 10 m by 10 m were required to intercept a sufficient number of bubbles for mass quantification per sample, and we intentionally collected samples with

different total gas volumes. We collected blank water samples to correct for background contributions of particulate matter, arsenic, lead, and chromium. Bubbling resulted in the visual accumulation of particles at the surface (Fig. S1) and in the sampling cup. During a separate field visit in November 2017 we used an Eckman dredge to collect sediment and stored the sediment in a 5 gallon bucket below 4 C° until use in February 2018 for bubble column experiments.

On 26 June 2018 we sampled for cyanobacteria bubble transport using similar procedures, except we used a simple

inverted funnel instead of a custom bubble size sensor to intercept rising bubbles. The sampling funnel was placed 10 m below the water surface, where cyanobacteria concentrations were expected to be lower than the surface based on previous observations (Preheim et al., 2016) to reduce sample contamination with cyanobacteria from the surrounding water column. Water temperature measurements taken using a Hydrolab sonde (Hach Co.) confirmed that the thermocline depth was above

10 m in this location during sampling. We collected 30-40 mL of water samples at 15m, 11m, 10m, and 1m depths for

background cell concentration counts, and gathered sediment grab samples with an Ekman dredge. All sample cups were

sterilized prior to use by rinsing with 10% bleach followed by 70% ethanol and deionized, sterile water, and cups were filled

with sterile water prior to sample collection. Samples were stored in a dark cooler on ice and were refrigerated upon return

to the lab. On 26 June 2018 we also used an Eckman dredge to collect a bulk sediment sample, which was kept in a dark

refrigerator at 4 C° until use in February 2019 for cyanobacteria transport in the experimental bubble column.


### 2.3 Large laboratory column design and sampling

To study bubble particle shedding and scavenging, we built a 15 m tall bubble column in the laboratory stairwell.

The column is comprised of four sections of 6-inch (15.3 cm) nominal diameter transparent polyvinyl chloride (PVC) pipe

joined by threaded unions with O-ring seals. The base of the column is a reducing tee fitting with a removable spigot for

drainage, and the column was filled from the top with tap water. We built a sediment container connected to 1/8 inch (3.1

mm) outer diameter copper tubing that could be lowered into the column and secured at any depth. The container was filled

with sediment originally collected with an Ekman dredge from the same place in UML used for field sampling. We used a

syringe pump to push air into the sediment through the tubing at a controlled rate, resulting in bubble release from the

sediment.

We conducted one set of column experiments in February 2018 to quantify shedding, scavenging, and metals

transport, and another set of column experiments in February 2019 to quantify cyanobacterial transport. For the shedding,

scavenging, and metals transport experimental runs, we filled the sediment bed with sediment collected from the same site as

ebullition experiments during our November 2017 field visit, and we injected 50 mL of air at 0.7 mL/min into the sediment

bed. Prior to the start of each run we collected water samples to correct for background contributions to particulate matter

and arsenic concentrations in bubble-transported particle data. Three experimental runs were each conducted at each of three

depths: 5m, 10m, and 15m, with the mobile sediment bed being repositioned between runs. To quantify particle scavenging

rates, we also conducted trials in which we injected air into the water column several centimetres above the sediment

surface. For the cyanobacterial transport experiments, we used sediment from the June 2018 field visit and injected variable

volumes of air into the sediment bed. We ran four experiments each at 6 m and 13 m depth, with the sediment being

replenished between the 6 m and 13 m runs. Six surface water grab samples were collected at multiple times throughout the

experiment to quantify background cell concentrations, and at each depth one trial was run where air was bubbled into the

water directly below sensor. For both sets of experiments, bubbles passed through the same customized bubble size sensor

(Delwiche et al., 2015;Delwiche and Hemond, 2017) and sample cup apparatus used in the field setting.

**2.4 Sample processing for particle mass and heavy metals analysis**

We filtered the field samples collected from UML for metals analysis within 24 hours of sampling with pre-

weighed Whatman Grade 41 quantitative cotton filters (nominal pore size 20 μm, 25 mm diameter). Due to filter clogging,

we typically used multiple filters for each sample. After filtering we air-dried the filters, weighed them, transferred each to

microwave digestion vessels, and added 10 mL of nitric acid from Fisher Scientific (Optima grade for ultra-trace elemental

analysis). Samples were digested in a MARS6 microwave oven, diluted with 30 mLs of Milli-Q water, and then filtered

with a 0.2 um polyethersulfone membrane syringe filter. For analysis, we diluted samples to 2% nitric acid, added a

rhodium internal standard, and analyzed the samples using an Agilent 7900 inductively coupled plasma mass spectrometer

(ICP-MS) with a 5 point calibration curve from 0.05 - 10 ppb. Blank analysis to determine background arsenic

concentrations in the Whatman cotton filter paper found levels at least two orders of magnitude below sample

concentrations. For metals analysis on bulk sediment sample we added 100 mg of dried sediment to 10mL of nitric acid and

digested as described above. The relative standard deviation values for counts-per-second from the ICP data were on

average 5.2% ± 2.8% for the bubble-transported sediment particles, and were 1.1% ± 0.6% for the bottom sediment digests

(which contained more particle mass per digest).

We filtered bubble column samples using pre-weighed 5.0 μm and 0.2 μm Whatman Nuclepore membrane filters

(47mm diameter) which use of only one filter of each pore diameter per sample. Filters were dried, weighed, digested,

diluted, and analyzed as described above. Blank analysis on Nuclepore membranes and lab filtering procedures found

arsenic contamination levels approximately two orders of magnitude below total arsenic concentrations in samples (total

being the sum of results from the 5 μmol filter and 0.2 μmol filter).

**2.5 Sample processing for cyanobacteria analysis**

For both the field and bubble column cyanobacterial transport experiments, we filtered a subset of the samples within 24 hours with 0.2 µm pore size filters held in autoclaved Swinnex filter holders (25 mm diameter). Filters were then removed from the filter holders and transferred to PowerWater bead beating tubes (Qiagen, Inc.). Approximately 8-9 mLs of remaining liquid for each sample was preserved with 1-2 mL of formamide (10% final concentration volume/volume) for microscopic cell counts. Lastly, the remaining sample volume was filtered on pre-weighed Whatman Nuclepore membrane filters (0.2 µm pore size, 47mm diameter), air dried, and re-weighed to estimate bulk mass transport.

For qPCR analysis on the June 2018 bulk sediment samples, 0.13 g of wet sediment was suspended in 15 mL of sterile water and then filtered as described above. For microscopy cell counts, 0.14 g wet sediment were preserved in 2% by volume paraformaldehyde. Water column samples from the June 2018 field campaign were also preserved in 2% by volume paraformaldehyde for cell counts. For qPCR analysis of the June 2018 sediment samples before use in the bubble column, we filtered 0.7 g of wet sediment (0.007 g dry sediment). For microscopy cell counts of the June 2018 sediment samples before use in the experimental bubble columns, we placed 0.8 and 2.0 mg of wet sediment (0.08 and 0.18 mg dry weight, respectively) in to 10 mLs of 10% formalin.

**2.6 Cyanobacteria cell quantification**

Cyanobacteria cell counts were assessed through quantitative polymerase chain reaction (qPCR) and microscopy. These two methods estimate cyanobacteria cells numbers by targeting different features of cyanobacterial cells. qPCR targets the unique genetic signatures in the 16S ribosomal RNA (rRNA) gene of cyanobacteria (Nubel et al., 1997) to estimate cell number from gene copy numbers. Microscopy takes advantage of the unique fluorescence spectra of cyanobacterial photosynthetic pigments to identify cells (Salonen et al., 1999). Positive control *Microcystis aeruginosa* UTEX LB 2386 and negative control *Pseudomonas aeruginosa* samples were used to optimize amplification conditions to ensure specificity for cyanobacteria qPCR. *Microcystis* and *Pseudomonas* cultures were grown overnight (12 h) under fluorescent lights at 25 °C in BG11 and Luria Broth media, respectively. *Microcystis* stock culture was serially diluted in phosphate buffered saline





to make a standard curve, filtered onto 0.22 μm polyethersulfone membrane filters (Millipore Sigma, Inc.) and frozen at – 80

°C until DNA extraction. Additionally, serial dilutions of *Microcystis* cultures were fixed with 1% formalin (final

concentration, volume/volume) for microscopy. While *Microcystis* cells were used as a positive control to test the method,

qPCR primers targeted all cyanobacteria cells (not limited to *Microcystis*).

To estimate the total number of cells in the *Microcystis* stock culture and samples with microscopy, between 4.6 mL

to 10.4 mL of fixed water samples or 1000 μL fixed *Microcystis* stock culture were filtered onto 0.22 μm polyethersulfone

membrane filters (Millipore Sigma, Inc.). Cells were visualized under a Zeiss AxioObserver Epifluorescence SIM

microscope [excitation: 545 nm; emission: 572 nm (Salonen et al., 1999)]. The total number of autofluorescent cells per

filter was estimated from twenty to forty random fields of view spanning the entire area of each filter. Cells were identified

from images with ImageJ (Schneider et al., 2012). First, background noise was reduced by excluding low intensity pixels,

with threshold values ranging between 14-162 (pixel intensities ranged from 0-255 for 8-bit gray-scale images). Next, only

particles within the size range of 0.1 μm$^2$ – 29.4 μm$^2$ were counted as cells. A dilution series of *Microcystis* fixed culture was

created by diluting cultures 2-fold in 1% formalin to test the variance and accuracy of this counting method (Fig. S3). We

did not test the quantification below 20 cells per field of view and all the experimental samples (not controls) had an average

of less than 20 cells per field of view, so microscopy measurements were only used for detection, not quantification.

For cyanobacteria cell quantification with (qPCR), DNA was extracted using PowerWater kits (Qiagen, Inc)

following the manufacturer's protocol, with the addition of 20 μl proteinase K and incubation at 65 °C for 10 min before

bead beating as an alternative lysis step. Primers were used to amplify *Cyanobacteria* 16S rRNA genes as previously

described (Nubel et al., 1997), with CYA359F (5'- GGG GAA TYT TCC GCA ATG GG) and an equal mixture of

CYA781R(a) (5'- GAC TAC TGG GGT ATC TAA TCC CAT T) and CYA781R(b) (5'- GAC TAC AGG GGT ATC TAA

TCC CTT T). qPCR reactions contained 10 μl of SsoAdvanced Universal SYBR Green Supermix (BioRad Laboratories,

Inc.), 1.6 μl DNA template, 2 μl forward primer (10 mM), 2 μl reverse primer (10 mM), and 4.4 μl deionized, reagent grade

sterile water. The following cycling conditions were used: denaturation at 98 °C for 30 seconds, annealing at 68 °C for 30

seconds, and elongation at 72 °C for 30 seconds followed by visualization step for 40 cycles. A dilution series of *Microcystis*

was created by diluting cells 10-fold in PBS before filtration and DNA extraction. Cell numbers for environmental samples





were determined from a linear regression of threshold cycle number (Cq) values of *Microcystis* and the number of cells

calculated for each dilution, (e.g. Fig. S4) and different batches were calibrated with internal standards of *Microcystis*

culture. Inhibition was determined for a subset of samples by spiking known concentrations of *Microcystis* DNA into

environmental DNA extracts and measuring the resulting threshold cycle number (Fig. S5). In all cases tested, inhibition was

negligible. The limit of quantification is 5 cells per filter, based on a signal to noise ratio (SNR) 2-3 x the average cell

concentration of the blanks (2.76 SNR).


**2.7 Cyanobacterial recruitment estimates for cells from Upper Mystic Lake**

        Values for recruitment estimates were calculated assuming ebullition occurs at an average rate of 22 mL m$^{-2}$ d$^{-1}$ for

the entire summer from all areas of the lake equally based on previous lake-wide ebullition surveys, (Varadharajan, 2009) .

We used the average cyanobacteria cell concentration from this study of 880 cells mL$^{-1}$ gas volume to calculate the average

flux of cells to the surface via ebullition ($C_e$) of 2 x 10$^4$ cells m$^{-2}$. We estimate the recruitment rate due to resuspension and

germination ($C_g$) as the maximum observed rate from a previous experiment at a different lake of 2.3 x 10$^5$ cell m$^{-2}$ d$^{-1}$

(Brunberg and Blomqvist, 2003), and applied this recruitment rate to areas of the lake suitable for germination, although this

is likely an overestimation.  The bounds for germination suitability from the sediments, found using previous measurements

of light, temperature, and oxygen at UML (Varadharajan, 2009), were assumed to occur 9 to 12 meters deep in the lake. The

fraction ($F_g$) of the surface area ($SA$ = 580,000 m$^2$) of lake above 9 and 12 meters that could support cyanobacterial

recruitment through germination is 0.40 to 0.50, respectively (Varadharajan, 2009). The contribution (%) of ebullition to

cyanobacteria recruitment ($P_e$) was calculated as:

$$P_e = 100 \times \frac{C_e \times SA}{(C_e \times SA) + (C_g \times F_g \times SA)} \qquad (1)$$

**3 Results and Discussion**


**3.1 Rate of bubble-particle transport**

        Both field and bubble column experiments demonstrate that bubbles transport particles from depths of at least 15 m

to the lake surface.  In field experiments, bubble-particle mass transport data showed a positive correlation between total



particle mass and gas volume in bubble traps for both field sampling campaigns (Fig. 1). This correlation was significant at

the $p < 0.05$ level for the October 2017 data ($r^2 = 0.76$), but only at the $p=0.15$ level ($r^2=0.38$) for the June 2018 data.  Total

gas collected per sample was a function of the number of anchor drops per sample and our ability to position the boat above

the bubble plume, so the lower gas volumes on June 2018 do not necessarily indicate a smaller reserve of gas in the

sediment.  Samples show variable mass transport rates both between and within sampling dates. Such variability is not

unexpected given the previously documented spatial and temporal variability of ebullition rates in this lake (Varadharajan,

2009;Varadharajan and Hemond, 2012).

Bubble column experiments show bubble particle transport from 15 m depth of $0.01 \pm 0.006$ mg/mL in the bubble

column, compared to $0.09 \pm 0.07$ mg/mL on October 2017 and $0.01 \pm 0.01$ mg/mL on June 2018 in the field.  This rate

reflects transport to the lake surface, but total bubble-particle transport rates would also include particles that were

transported part-way up the bubble column and then shed from the bubble.  To estimate the significance of particle shedding,

we used the bubble column to compare transport rates from bubbles released at 5 m, 10 m, and 15 m depths (Fig. 2).  We

found no significant difference in transport rates, indicating either that particle shedding was not significant, or that

scavenging of particles from the water column equalled particle shedding from bubbles.  We did however note that the first

bubble column test conducted after repositioning the sediment source yielded a higher particle transport rate than those

found in subsequent tests (Fig. 2), which may be attributable to some sediment disturbance during movement of the sediment

source. We also note that while bubbles do dissolve as they rise, bubbles in the size range seen during this study remain

relatively constant over 15 m because dissolution is partially compensated by bubble expansion during rise (Delwiche and

Hemond, 2017), and we therefore do not expect bubble dissolution to substantively impact particle shedding.

To quantify the role of bubble scavenging of particles from the water column, we compared the above data from 5

m and 10 m column experiments to samples gathered when gas was bubbled from several centimetres above sediment, thus

allowing maximum opportunity for scavenging to occur.  Particle mass scavenging represented only around 10% of the mean

particle transport for 5 m and 10 m experiments (grey diamonds in Fig, 2), indicating that while scavenging rates were non-

zero in the column, the large majority of the particulate matter transported to the top of the water column originated in the

sediment.  The lack of significant bubble particle scavenging provides validation for our field methodology, because any

sediment that is artificially stirred up by the anchor should not have an appreciable effect on particle transport rates.

Furthermore, the relative lack of particle shedding or scavenging indicates that the large majority of particles deposited on

the surface by bubbles originate in the sediment.

While bubbles transported sediment directly from the bottom of the laboratory column to the water surface, a

vertical distance of 15 m, there appears to be no reason that transport of particles from significantly larger depths cannot

occur. Such transport provides a direct chemical and biological link between sediment and surface waters, and this could be

the dominant link between deep sediments and the surface water during months of stratification.

**3.2 Bubble size distribution similar between field triggered and natural bubbles**

Bubble volume has been found to significantly affect particle flotation rates in industrial processes (Yoon and

Luttrell, 1989), and therefore it is important to compare the anchor-triggered bubble sizes to naturally-occurring bubble sizes

to understand how our measured transport rates may reflect naturally occurring transport rates. Anchor-triggered bubbles

were significantly smaller (average diameter 5.6 mm) than those measured for natural bubbling events (average diameter 6.4

mm) during a 2016 field campaign [Fig. S6, (Delwiche and Hemond, 2017)]. However, both natural and triggered bubbles

were still very large compared to bubbles used in traditional flotation chambers (Yoon and Luttrell, 1989;Rubio et al., 2002).

While research on particle flotation for large bubbles is limited, several previous studies have found that differences in

particle transport rates decrease for bubbles above 1 mm diameter (Dai et al., 1998;Koh and Schwarz, 2008), indicating that

particle transport rates should be similar between natural and triggered bubbles. Bubble sizes measured in the cyanobacteria

transport experiment displayed a bimodal distribution (Fig. S7) that was not observed in other bubble experiments. This

bimodal distribution could be a result of artificially pumping gas in to the sediment, but the impact of this on particle

transport is unknown.


**3.3 Chemical and biological evidence indicates that particles originate in the sediment**

The data on bubble particle mass transport clearly shows that bubbles are capable of transporting particles from

relatively deep depths, and minimal rates of particle shedding and scavenging in the water column indicates that these



particles originate in the sediment. Concentrations of arsenic, chromium, and lead in the bubble-transported particulate

matter collected during field experiments were similar to concentrations in the sediment (Fig. 3). Bubble- transported

particles contain arsenic, chromium, and lead at average ratios of 100 µg kg$^{-1}$ , 120 µg kg$^{-1}$ , and 240 µg kg$^{-1}$ (respectively,

excluding outlier in chromium data, see Fig 3), compared to 136 ug kg$^{-1}$, 160 ug kg$^{-1}$, and 330 ug kg$^{-1}$ (respectively) in bulk

sediment samples. In the bubble column, arsenic and chromium levels are similar to the bulk sediment in the column

experiments (Fig. S8), although lead levels appear to be higher. Overall, this similarity supports our conclusion that bubbles

are primarily transporting sediment matter to the lake surface with only modest amounts of scavenging or particle shedding,

despite the relatively deep water column.

In addition to the heavy metal results indicating that the transported particles are from the sediment, biological

evidence also suggests a sedimentary origin. All particle samples transported by bubbles contained an abundance of

biological structures (Fig. S9), such as the apparent head shields and carapaces of *Bosmina spp.*, which have also been found

extensively in other freshwater lake sediments (Kerfoot, 1995). These particle samples also contained structures that appear

to be ephippia, the protective cases enclosing diapausing eggs produced by zooplankton such as *Daphnia*. Ephippia can

overwinter in lake sediments or survive periods of desiccation, providing a seed bank to recolonize the water column when

favorable conditions return (Caceres and Tessier, 2003;Hairston, 1996). These biological findings further support the

finding that bubbles are transporting sediment particles through the profundal water column.

**3.4 Implications for arsenic and heavy metal cycling**

The presence of arsenic and other heavy metals in the bubble-transported particles could have significant

implications for chemical cycling in aquatic ecosystems. Measured rates of arsenic flotation in field samples were about 8 ±

4  µg arsenic L$^{-1}$ of gas bubbled, about 12 ± 7 µg L$^{-1}$ for chromium, and 22 ± 12 µg L$^{-1}$ for lead (Fig. 4). Typical natural

bubble flux for UML was estimated as 0.022 ± 0.020 L m$^{-2}$ day$^{-1}$ during previous ebullition studies (Varadharajan, 2009),

which corresponds to an estimated arsenic flux of 0.18 ± 0.19 µg m$^{-2}$ day$^{-1}$ from the sediment to the lake surface. This flux

would be highly episodic given the spatial and temporal heterogeneity of methane bubbling in UML (Varadharajan,

2009;Scandella et al., 2016).

This estimate of daily arsenic flux can be compared with historical measurements related to arsenic cycling in UML (UML has not been extensively studied for chromium and lead cycling). In 2000, for example, the arsenic accumulation rate

in the epilimnion of UML was approximately 38 µg m$^{-2}$ day$^{-1}$ (Knauer et al., 2000). This flux is two orders of magnitude larger than our estimate for bubble transported arsenic of 0.18 µg m$^{-2}$ day$^{-1}$, indicating that bubble-arsenic transport may be of relatively low importance in UML where a significant fraction of the arsenic input to epilimnetic waters can be attributed to inflow from the Aberjona River (Hemond, 1995). However, bubble-mediated fluxes of arsenic or other sediment-borne metals may represent a larger fraction of epilimnetic input in other lakes having lower influx rates from surface water inflow.

In addition, particles transported by bubbles are deposited directly at the water surface, potentially positioning them for easy ingestion by recreational swimmers.

Although bubble-facilitated transport does not appear to dominate arsenic transport in UML, much higher ebullition rates have been reported elsewhere in the world (Deemer et al., 2016). For example, a mid-latitude reservoir in Switzerland was reported to have an order of magnitude higher ebullition flux [0.225 L m$^{-2}$ day$^{-1}$ (Delsontro et al., 2010)]. Co-occurrence

of high ebullition rates and contaminated sediment could lead to significant bubble-facilitated contaminant cycling.

**3.5 Implications for cyanobacteria transport and possible bloom initiation**

Since cyanobacteria are known to overwinter in lake sediments, bubble-mediated transport could be one mechanism by which resting cells inoculate the upper water column. We estimated the concentration of cyanobacteria cells in sediments

and associated with bubble-transported particles with quantitative polymerase chain reaction (qPCR), and then used bubble column experiments to demonstrate cyanobacteria cell transport via bubble flotation. The column was filled with tap water to minimize cyanobacteria contamination from the surrounding water column. Bubbles were emitted directly under the bubble trap as a control, or within a sediment bed placed at 6 or 13 m below the surface. Prior to bubbling, the water column cyanobacteria cell concentration was less than 1 cell mL$^{-1}$ of water (Fig. 5). After bubbling air through the sediment at the

bottom of the column, the cyanobacteria concentration in the column increased to an average of approximately 7 cells mL$^{-1}$, but still remained lower than the concentration of cells in the trap (Fig. 5). Bubble-transported particulate matter contained cells at a rate of approximately 30 cells mL$^{-1}$ gas, indicating that bubbles are capable of transporting cyanobacteria through





deep water columns.  We also measured cyanobacteria transport in the field with bubble traps, but our measurements were contaminated by cyanobacteria in the surrounding water column (see SI for results and discussion).

Since bubble particle transport rates varied across experiments (from $0.01 \pm 0.006$ mg/mL in the bubble column to $0.09 \pm 0.07$ mg/mL on October 2017 and $0.01 \pm 0.01$ mg/mL on June 2018 in the field), and cell flotation rates are likely tied to total particle transport rates, we conclude that the best estimates of transport of cyanobacteria cells in the field should be based on average estimates of particle transport per L gas across all column and field experiments and the average cyanobacteria cell concentration in the sediment.  From an average sediment transport rate across all field and column

experiments and an average sediment concentration, we can extrapolate that approximately $8.8 \times 10^5 \pm 11.4 \times 10^5$ cyanobacteria cells are transported per L bubble volume in the lake.

To determine the importance of bubble-mediated transport in inoculating surface waters with cyanobacteria cells from the sediment, we use this transport estimate of $8.8 \times 10^5 \pm 11.4 \times 10^5$ cells L$^{-1}$ and the bubble flux estimate mentioned previously of $22 \pm 20$ mL m$^{-2}$ day$^{-1}$ to estimate a daily transport of $1.9 \times 10^4 \pm 3.1 \times 10^4$ cells m$^{-2}$ day$^{-1}$. Harmful

cyanobacterial species, such as *Microcystis*, often aggregate in the surface layers to outcompete other algae for sunlight (Xiao et al., 2018). If cyanobacteria cells concentrate within the upper 1 meter of the lake, this results in an approximate increase in concentration of 20 cells L$^{-1}$ day$^{-1}$. While this concentration is not a human health concern, such a concentration from the average rate of bubbling could represent a significant inoculum. At a maximum growth rate of approximately 1 day$^{-1}$ (Robarts and Zohary, 1987) and absent significant losses, this cell concentration alone would result in a cell density greater

than the Massachusetts Health limit of $7 \times 10^4$ cells mL$^{-1}$ in about three weeks. Larger bubbling events [e.g. (Deemer et al., 2016)] could result in the same cyanobacteria cell concentration within approximately 15 days. The estimated growth of bubble-transported cyanobacterial cells is dependent on the cells being viable.  Incubating cells to assess viability will be an important step for future studies.

Bubble-mediated transport could be a significant mechanism of recruitment of cells to the surface waters especially

in deep, stratified lakes like UML. Cyanobacteria are thought to largely be recruited to surface waters from shallow areas due to a combination of higher light, temperature, and oxygen levels that promote germination, and increased wind-driven sediment resuspension (Ramm et al., 2017). While sediment cyanobacteria concentrations are higher in deeper areas of the

lake, cells are not able to germinate because of the dark, anoxic conditions in deep, eutrophic lakes (Ramm et al., 2017). Bubble-mediated transport is a mechanism by which this large reservoir of "lost" cells in deep sediments could contribute to

overall recruitment to the surface waters. To determine the potential contribution of bubble mediated transport to cyanobacteria recruitment to the surface, we assume that germination does not occur significantly past the oxycline (9 - 12 m) in UML between June and Oct., as low oxygen concentrations and low light levels prevent germination, and wind-driven mixing cannot resuspend sediments across the shallow thermocline (Varadharajan, 2009). We also assume that cells resuspended in the spring overturn in March would have germinated, settled, lysed or have been consumed by grazers by

June [e.g. (Tijdens et al., 2008;Verspagen et al., 2005)]. Furthermore, we do not include external inputs of cyanobacteria to the lake, such as from the river [e.g. (Bouma-Gregson et al., 2019)] or air (Seifried et al., 2015;Lewandowska et al., 2017;Evans et al., 2019). Since literature estimates of recruitment rates for these sources are lacking, we assume these inputs are small compared to shallow sediment and bubble-mediated recruitment. Using the maximum observed recruitment rate of $2.3 \times 10^5$ cells m$^{-2}$ day$^{-1}$ (Brunberg and Blomqvist, 2003) from sediments for the area of the lake above 9-12 meters, we

estimate that bubbling could contribute to between 14 - 17% of cyanobacterial recruitment in the lake.

**4 Conclusions**

Bubble-particle transport between the sediment and surface of UML is a novel transport pathway capable of moving particulate matter upwards through a stratified water column, over depths of 15 m or greater, without shedding a major

fraction of their particle burden or accumulating large amounts of additional particles as they rise. Bubble-facilitated metal transport in present-day UML appears minor compared to surface inflows, but lakes with higher ebullition flux or more contaminated surficial sediments may experience more significant chemical transport from contaminated sediments. Bubble mediated transport of cyanobacteria cells may contribute significantly (approximately 15% of the total in UML) to cellular recruitment from the sediment, thereby inoculating epilimnetic waters of stratified water bodies with cyanobacterial cells, at

rates capable of triggering significant blooms in a few weeks. Bubble transport is expected to be particularly important in deep, eutrophic lakes in which alternative mechanisms of sediment regeneration to surface waters are limited. Further work is warranted to more thoroughly quantify this ebullitive transport pathway, and its implications for chemical and biological cycling. In addition, future work should include alternative methods of bubble triggering as well as the quantification of particle transport rates on naturally-occurring bubbles.


**Code/Data availability**





All data necessary to validate the research findings are available on JHU Dataverse, doi.org/10.7281/T1/7WXPIN.

**Author contributions**

KD, HH and SP designed the experiments and KD and JG carried out experimental work. KD, JG and SP analyzed the data.

KD prepared the manuscript with contributions from all co-authors.

**Competing interests**

The authors declare that they have no conflict of interest.

**Acknowledgements**

Funding for part of this work was supported by Johns Hopkins Whiting School of Engineering. Funding was also provided by the Singapore-MIT Alliance for Research and Technology, MIT Center for Environmental Health Science, the MIT Martin Family Fellowship to K. Delwiche, the W. E. Leonhard 1941 professorship to H. Hemond, and the National Science Foundation under grant number EAR-1045193. This work was also supported in part by the NIEHS Superfund Basic Research Program, NIH, P42 ES027707.

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

**Figures and Legends**

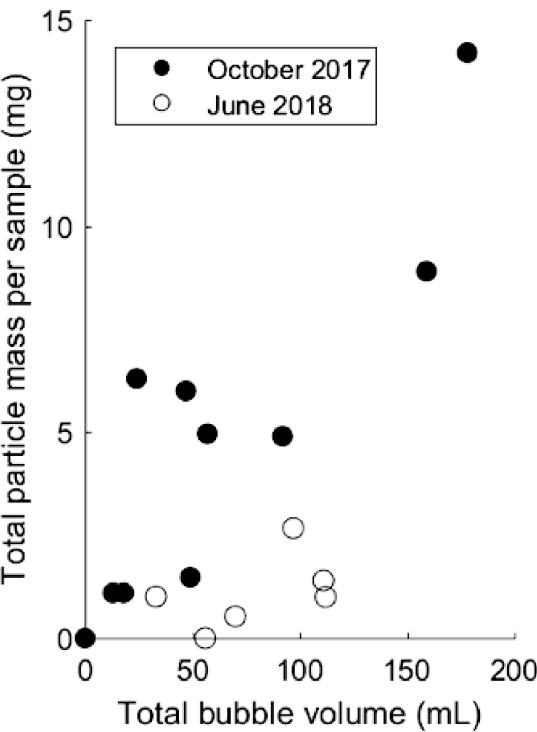

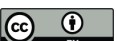



**Figure 1:** Total particle mass in mg associated with the bubbles captured during each field campaign with bubble triggering events in Oct. 2017 (filled circles) and June 2018 (open circles). Triggering events yielded different bubble volumes (given in mL).

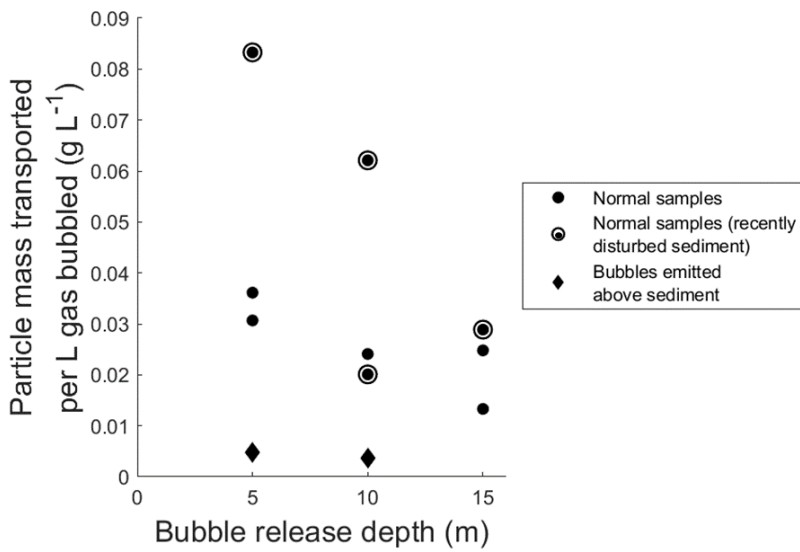


**Figure 2:** Transported particle mass per L of gas bubbled in the large bubble column, as a function of bubble release depth. Solid circles represent samples where bubbles were emitted from the sediment bed, diamonds represent samples where gas was bubbled directly above the sediment bed. Hollow circles around solid circles denote samples with recently-disturbed sediments.






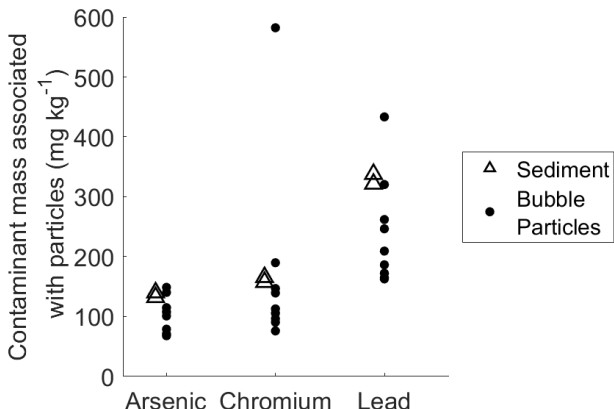

**Figure 3:** Comparison between mass of arsenic, chromium, and lead per kg of sediment (open triangles) and bubble-transported particulate matter (solid circles). Standard deviation scale similar to point size and therefore omitted for figure

clarity.

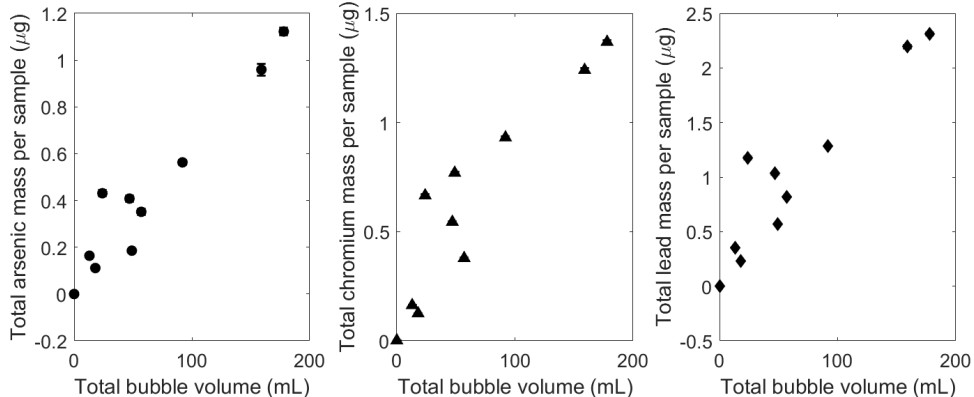

**Figure 4:** Chemical amounts observed in bubble traps associated with bubble-mediated transport of sediment particles. (a)

Arsenic mass, (b) chromium mass, (c) lead mass (in µg) transported versus the bubble volume of each sample (in mL, as

measured at the lake surface a-c). Standard deviation is added to each measurement but the scale similar to point size for

most measurements.




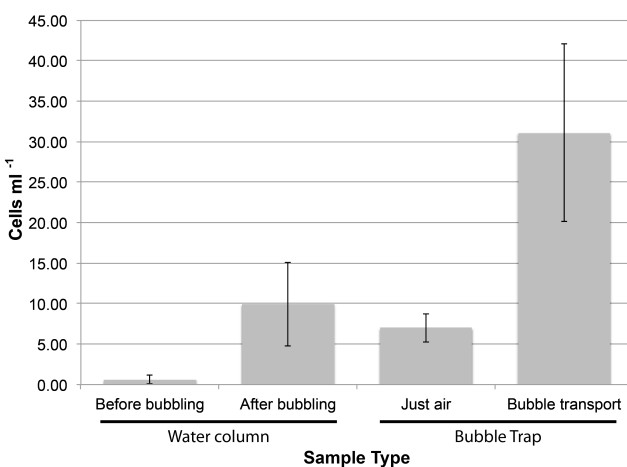

**Figure 5:** The concentration of cyanobacteria cells (as measured by quantitative PCR) increases in the experimental water column and bubble traps after initiating bubbling within sediments. The background concentration of cyanobacteria cells in water in the column was initially very low (Before bubbling) but increased after bubbling. The concentration of cells in the bubble trap increased because of the contamination from the surrounding water column, even without bubbling within sediment (Just air). However, the highest concentration of cyanobacteria in the bubble trap was observed when initiating bubbling from within the sediment (Bubble transport). The increase in cell concentration in both the water column and the bubble trap after bubbling within sediment is evidence for cyanobacteria transport via bubble floatation. Error bars show standard deviation across measurements.