# Peer review of "Vertical transport of sediment-associated metals and"

_Biogeosciences, 2019_

## Referee Comment (RC1) · Dileep Kumar (Referee) · 12 Jan 2020

Transfer of substances across interfaces assumes significance to understanding material balances, drivers and functioning of ecosystems. In aquatic systems transport of materials from sediments to overlying water column is known to occur through diffusive fluxes and episodically under sediment disturbed conditions. The role of bubbles is well recognized as transporters of materials from water column to surface layers but the current manuscript adds a new dimension by demonstrating export of sediment particles to surface and quantifying associated substances including biota transfer to upper layers by this mechanism. The results are significant with particular reference to

shallow aquatic ecosystems and also highlight the role of bubbles in material exports in other in systems of relevance. In general the manuscript is presented well but requires some minor revisions as per the following observations:

Abstract

Line 9: 'Bubbles adsorb and transport particulate matter both in industrial and marine systems' – include lakes systems here.

Line 9-12: 'methane-containing bubbles emitted from anoxic sediments are found extensively in aquatic ecosystems' – the word "extensively" is inappropriate for marine systems since methane- containing bubbles can only be found in a few select coastal ecosystems. However, this issue assumes greater and global significance in vertical transportation of dissolved and particulate materials scavenged across a few meters below to sea surface by the rising wind-induced bubbles, particularly in shallow marginal systems.

Introduction

Lines 34-35: 'Metals can be mobilized from sediments via solubilization by oxidation-reduction reactions, and by sediment resuspension or bioturbation' – mobilization can also occur through acidification of lakes.

Line 35-36: 'transport to surface waters of contaminants mobilized from the sediment is affected by lake hydrodynamic conditions, notably stratification' – an interesting question to the current investigation be how does stratification influence methane bubble rising to surface during minimal wind induced turbulent conditions? A strongly stratified upper water column will inhibit (slow down speed of rising) or even prevent particularly small sized but proportionately with large surface areas from rising across the strong pycnocline. This is possible if vertical profiling is done with close intervals of sampling to find density gradients across the pycnocline and assessing the bubble rise rates in hypo- and epilimnion layers.

Lines 40-41: 'Verspagen et. al. (2005) showed that recruitment from sediments of the potentially toxic cyanobacterium Microcystis was a major driver of the summer bloom (Verspagen et al., 2005)' – referenced twice in the same sentence!

Lines 66-67: 'the full extent of bubble particle flotation in aquatic systems remains unknown.' – even the present manuscript cannot make it 'full', which requires many attempts by many investigators!

Line 70: Fig. S1 should show pictures before and after the bubble event to highlighting the emergence of particles following the bubbling.

Lines 78-80: 'Given the expected importance of both bubble size and total bubble volume, we used a bubble size sensor (Delwiche et al., 2015; Delwiche and Hemond, 2017) to measure bubble diameter distributions both in the lake and in the laboratory.' – adsorption or scavenging of particles by bubbles is expected to be proportional to the surface area of the bubble (similar to metal adsorption on to a particle) and therefore representing bubble characteristic in terms of 'surface area' than its 'size' or 'volume' would be preferable.

Methods

Lines 101 and 250: 'another lake' – please name the lakes.

Lines 111-112: 'All bubbles rising through the bubble size sensor or collection funnel entered the flexible tubing and rose into the sample cup.' – as the particles and the associated substances are adsorptive in nature it is likely that some of the rising bubble attached particles are adsorbed in the flexible tubing etc. before they reached sample cup. Authors may include a statement on this possible loss of particles during sample processing.

Line 117: Word 'approx.' may not be necessary as the coordinates are specified to third decimal.

Lines 119-120: 'preventing mixing from of the sediment to the surface.' – requires

rephrasing.

Lines 121-124: Good strategy.

Line 250: Please correct the flux units 'cells m-2' to cells m-2 d-1.

Results and Discussion

Lines 262-263: 'demonstrate that bubbles transport particles from depths of at least 15 m to the lake surface.' – It may be revised as "demonstrate that bubbles transport particles from depths to the lake surface" since bubbles if formed even in deeper waters can transport materials to surface.

Line 307: Lines 134-135 mention 'On 26 June 2018 we sampled for cyanobacteria bubble transport using similar procedures, except we used a simple inverted funnel instead of a custom bubble size sensor to intercept rising bubbles' whereas Fig. S7 caption shows "Frequency distribution numbers are approximate because the bubble size sensor is unable to measure fast bubble flux or very small bubbles" – It is important to check the compatibility between these statements, particularly for data of 26 June 2018 if used.

Line 317: replace ug with $\mu$g.

Lines 342-343: Besides 'a significant fraction of the arsenic input to epilimnetic waters can be attributed to inflow from the Aberjona River (Hemond, 1995)' aerial transport of dust associated arsenic/metals should be invoked here to be among the unknown inputs.

Lines 361-362: 'Bubble-transported particulate matter contained cells at a rate of approximately 30 cells mL-1 gas, indicating that bubbles are capable of transporting cyanobacteria through' – May be revised as "Bubble-transported particulate matter contained cells at approximately 30 cells mL-1 gas, indicating that bubbles are capable of transporting cyanobacteria through". A 'rate' is expected to be material transferred during a specific duration (time). ***

---

## Referee Comment (RC2) · Anonymous Referee #1 · 24 Jan 2020

Delwiche and others quantify the particle transport by bubbles released from the lake sediments. Although interesting, I found several flaws in both the field and laboratory set-up, suggesting a severe limitation of the work.

• Authors state that the particles associated with the bubbles, almost entirely originated from the sediments, rather than from the water. Will this statement hold true in case of turbid waters? Please clarify.

• Add the details of dissolved oxygen concentration, temperature and total suspended matter in the water column at the lake sampling station.

• Did you observe any bubble breakup during the transport through the flexible tub-

ing? If yes, does it affect the final bubble size count and volume transported?

• Line 114, please add the grade of HNO3 used for rinsing.

• Authors dropped a cinderblock to trigger bubble release. Please state the difference in bubble volume during natural release and forced release.

• The impact of cinderblock on the lake floor would have re-suspended a significant amount of sediments. Does the forced release, thus suggest a much larger than natural bubble release mediated particle transport?

• The collection of sediment by dredge and subsequent transport in bucket, would have resulted in the release of a significant amount of gas from the sediments. Can the authors provide the difference in the gas content of in-situ sediments and those collected by dredge and brought to the lab in a bucket?

• What was the percentage of bubbles breaking up, when striking the inverted funnel and releasing the cyanobacteria?

• Authors used air, instead of methane in the laboratory experiment. Will there be a difference in the particle transport by an air bubble as compared to methane bubble? Please discus in the text.

• How did the authors decide the rate of injection of air into the sediments? What happened to the gases already present in the sediments when authors injected the air?

• Line 266, authors did not estimate the gas reserve in the sediments. How can they infer that the lower gas volume did not indicate a smaller gas reserve?

• If the positing of boat influenced the bubble release, then how can they quantify the bubble volume and associated particle transport?

• Line 273, I do not agree with the comparison of experimental column release with that from the natural lake environment. As stated above the conditions in the lab were completely different than that in the lake, and thus any comparison between the two is

superfluous.

• Authors state a large difference in the size of natural and forced release of bubbles. Then what is the reliability of the volume and particle transport estimated by the authors?

Line 25, change 'Concentrations' to 'Concentration'

Line 27, change 'concentrations' to 'concentration'

Line 40, modify 'et. al.' with 'et. al.'

Line 48, insert space after 2008;

Line 71, change 'volumes' to 'volume'

Line 74, change 'greatest' to 'a considerable'

Line 79, change 'distribution' to 'distribution'

Line 119, change 'mixing from of the' to 'mixing from the'

Line 123, change 'an' to 'a'

Line 148, change 'column is comprised' to 'column comprised'

Line 176, change 'um' to '$\mu$m'

Line 180, change 'metals analysis on bulk sediment' to 'metal analysis in bulk sediment'

Line 185, change 'which use' to 'with use'

Line 186, change 'analysis on' to 'analysis of' Line 188, 5 $\mu$mol filter? Is it correct?

---

## Referee Comment (RC3) · Anonymous Referee #4 · 27 Jan 2020

General Comments: The manuscript by Delwiche et al describes the contribution of bubble-mediated, vertical transport of particles, chemistry, and biology within a lake. This is a valuable observation with a unique laboratory experiment to support the results. The observations found here are fascinating and worthy of publication. I am supportive of publication; however, the issues with sample collection make me call into question the quantitative results and budget. Please see my specific comments below for further details. Ultimately, the data need to be published, but the manuscript needs major revisions to remove the budgets which are likely inaccurate, given the sample collection procedure. Please refocus the manuscript to state the observations and cast your results in light of how the samples were collected.

[Figure]

Specific Comments: L 23-24: Define "problematic". What does this mean for cyanobacteria? Be more specific. L 29-30: What about the "improved understanding"? What type of understanding? Be specific. L 110-111: How do you know the bubble-transported biology and chemistry is no adhered to the inner walls of sampling equipment? Do your measurements represent an underestimate? L 172-173: Are these filter measurements meant to be volumetric? If so, do you know how much water passed through each filter before clogging? L181: I don't know how this relates to the accuracy and precision of your measurements? How do counts per second relate to concentration? L 266: This is an excellent study and I think your experiments and testing shows bubbles play a role in lakes that has not been considering from a biological perspective. This study needs to be published, but I can't get over the anchor drop issue. I have thrown many anchors overboard in lakes and the plume of sediment is always significant. I have a hard time decoupling this disturbance with your results. There needs to be a paragraph describing how the laboratory results follow the lake results and the anchor had minimal impact on the lake results. Although, your laboratory results show sediment disturbance impact the bubble transported particles. How can you decouple these methodological problems with your results? What if you shift the focus of your manuscript to documenting that bubbles DO transport chemistry and biology, but stop short of the full budgets, as I think those are biased due to the methodological problems. L 268-270: This observation is baseless since you caused the ebullition. L 277-280: This is analogous to dropping an anchor on the lake sediments. How do you reconcile these laboratory experiments with what you did in the field? Again, this is evidence the focus of the manuscript should be focused to an observation that bubbles do transport chemistry and biology, but do not calculate budgets because the evidence shows they are not accurate. L 283-285: Were there particles to scavenge? This was tap water, right? Section 3.3 header: Again, I have a hard time reconcile the topic of this section that particles originated in the sediment after traveling through a plume of sediment. Maybe scavenging is a more active process and makes up a larger percentage of the particles when not passed through a plume of sediment. L 325-326:

Observations like this are the reason this manuscript needs to be published. L 353-354: This is a major finding of this study and should be a highlight. L 374: What does it mean to have a negative rate of transport? Are bubbles actually sequestering cells from the surface waters? This is another reason why I think the budgets need to be removed and the focus placed on the observations and laboratory experiments. L 400: Given the large errors in your bubble transport of cells, I have a hard time following how the error now is so small. The error propagation is not well explained.

Technical Comments: L 22: Delete "are". L 22-23: First sentence needs a citation. L 32-34: First sentence of the paragraph, poor sentence structure, please rewrite. L 35-37: "However, transport to surface..." Poor sentence structure, please rewrite. L 45-46: "Bubbling from anoxic sediment..." Sentence missing numerous citations. L 50-53: "Bubble-mediated particle..." Poor sentence structure, confusing, please rewrite. L 184-185: "We filtered bubble..." I did not understand this sentence. L 187: How much lower are the blanks? Actual numbers would be better. Two orders of magnitude can range from 110-fold lower to 900-fold lower. These are very different blanks. L 249: mL-1 gas volume or mL gas volume-1? L 250: Estimate – estimated (past tense). L 258: Bring eq. 1 up so that the reader knows the equation before getting the variables. Rewrite the part about the depth interval for germination. I was lost. L 362: This is a concentration, not a rate. L 365: Keep units consistent. Use slash or exponent throughout.

---

## Author Comment (AC1) · 17 Feb 2020

*We would like to thank Dr. Dileep Kumar for looking over this work and providing valuable critiques to our paper. The comments are thoughtful and bring up many important points, which we addressed individually below.*

Abstract Line 9: 'Bubbles adsorb and transport particulate matter both in industrial and marine systems' – include lakes systems here.

*We agree that this opening sentence falls short of encompassing the full scope of the impact of bubbles. Industrial and marine systems have been the primary focus of the*

[Figure]

*research, but this phenomenon applies more broadly, and certainly applies to lakes as we show in the manuscript. We adjusted the opening sentence to provide a broader significance to the work.*

*"Bubbles adsorb and transport particulate matter **in a variety of natural and engi- neered settings, including** industrial**, freshwater,** and marine systems."*

Line 9-12: 'methane-containing bubbles emitted from anoxic sediments are found ex- tensively in aquatic ecosystems' – the word "extensively" is inappropriate for marine systems since methane- containing bubbles can only be found in a few select coastal ecosystems. However, this issue assumes greater and global significance in verti- cal transportation of dissolved and particulate materials scavenged across a few me- ters below to sea surface by the rising wind-induced bubbles, particularly in shallow marginal systems.

*Thank you for this comment. We changed the sentence to reflect that methane contain- ing bubbles would be a particular concern in freshwater systems such as lakes, reser- voirs, and wetlands ("are found **widely** in **freshwater** ecosystems"), and also broad- ened the first sentence to bring in more of the global significance of bubble-mediated transport (above).*

Introduction Lines 34-35: 'Metals can be mobilized from sediments via solubilization by oxidation reduction reactions, and by sediment resuspension or bioturbation' – mobi- lization can also occur through acidification of lakes.

*Thank you for bringing this omission to our attention. We have added acidification to the list of mechanisms.*

*"mobilized from sediments via solubilization by oxidation-reduction reactions, and by sediment resuspension, **acidification,** or bioturbation (Calmano et al., 1993;Eggleton and Thomas, 2004;Schaller, 2014; **Schindler et al., 1980**). "*

Line 35-36: 'transport to surface waters of contaminants mobilized from the sediment is

affected by lake hydrodynamic conditions, notably stratification' – an interesting question to the current investigation be how does stratification influence methane bubble rising to surface during minimal wind induced turbulent conditions? A strongly stratified upper water column will inhibit (slow down speed of rising) or even prevent particularly small sized but proportionately with large surface areas from rising across the strong pycnocline. This is possible if vertical profiling is done with close intervals of sampling to find density gradients across the pycnocline and assessing the bubble rise rates in hypo- and epilimnion layers.

*The impact of stratification on bubble rise is an interesting question to both methane-emission from lakes and bacterial or chemical transport that should be addressed in future work. If stratification does prevent small bubbles from penetrating, bubbles may be an additional mechanism concentrating organisms or chemicals at these interfaces, resulting in the thin layers of organisms that can often be seen within the water column. However, the changes in density across naturally occurring pycnoclines might be gradual enough and the bubbles buoyant enough pass through it without bursting or accumulating. The proposed vertical profiling experiment would make a very good follow-up study.*

*We have also added a discussion of these questions "**However, many questions remain regarding bubble-mediated transport in natural systems, including how the change in water density at the thermocline affects bubble rise and associated chemical and biological material.**"*

Lines 40-41: 'Verspagen et. al. (2005) showed that recruitment from sediments of the potentially toxic cyanobacterium Microcystis was a major driver of the summer bloom Verspagen et al., 2005)' – referenced twice in the same sentence!

*Thank you for pointing out this redundancy. It has been changed to "**Previous research** showed .."*

Lines 66-67: 'the full extent of bubble particle flotation in aquatic systems remains

unknown.' – even the present manuscript cannot make it 'full', which requires many attempts by many investigators!

*We agree we cannot hope to determine the full extent of bubble particle flotation with this study, and have removed full from this sentence.*

Line 70: Fig. S1 should show pictures before and after the bubble event to highlighting the emergence of particles following the bubbling.

*We have changed Figure S1 to show the water surface near the beginning of a bubble triggering event, as well as at the end. This highlights the visible accumulation of particulate matter on the water surface.*

Lines 78-80: 'Given the expected importance of both bubble size and total bubble volume, we used a bubble size sensor (Delwiche et al., 2015; Delwiche and Hemond, 2017) to measure bubble diameter distributions both in the lake and in the laboratory.'– adsorption or scavenging of particles by bubbles is expected to be proportional to the surface area of the bubble (similar to metal adsorption on to a particle) and therefore representing bubble characteristic in terms of 'surface area' than its 'size' or 'volume' would be preferable.

*The surface area is an important bubble characteristic for transport along with other key characteristics, but typically the bubble diameter is provided as a key metric of bubble characteristics. We have rephrased this as. "Given the expected importance of bubble **size on key characteristics (e.g. surface area, buoyancy, diffusion of gas)**, we used a bubble size sensor (Delwiche et al., 2015;Delwiche and Hemond, 2017) to measure bubble diameter distribution both in the lake and in the laboratory."*

Methods Lines 101 and 250: 'another lake' – please name the lakes.

*The lakes referenced as "other lakes" in the text are **Lake ScharmuÌ́Ltzelsee and Lake Limmaren**, which have been explicitly stated in the text.*

Lines 111-112: 'All bubbles rising through the bubble size sensor or collection funnel

entered the flexible tubing and rose into the sample cup.' – as the particles and the associated substances are adsorptive in nature it is likely that some of the rising bubble attached particles are adsorbed in the flexible tubing etc. before they reached sample cup. Authors may include a statement on this possible loss of particles during sample processing.

*The reviewer makes a valid point, transported particles were indeed adsorptive and some stuck to the sample tubing. We have added a statement to introduce this possible sample processing artifact.* "**The interaction of bubbles with the flexible tubing resulted in visible particle attachment to the tubing, making our estimates of particle mass transport a lower bound**"

Line 117: Word 'approx.' may not be necessary as the coordinates are specified to third decimal.

*This was removed.*

Lines 119-120: 'preventing mixing from of the sediment to the surface.' – requires rephrasing.

*We have rephrased as "preventing mixing **of** sediment to the surface"*

Lines 121-124: Good strategy.

*We found sediment contamination within the collection cups that were deployed for much longer periods capturing natural bubble events, but the possibility of contamination and subsequent growth, death or decay of transported cells made it impossible to have reliable estimates from this method. While there are some issues with this approach, it made the measurements of particle transport feasible.*

Line 250: Please correct the flux units 'cells m-2' to cells m-2 d-1.

*This error was corrected.*

Results and Discussion Lines 262-263: 'demonstrate that bubbles transport particles

from depths of at least 15 m to the lake surface.' – It may be revised as "demonstrate that bubbles transport particles from depths to the lake surface" since bubbles if formed even in deeper waters can transport materials to surface.

*This statement is confusing, and was revised to " Both field and bubble column experiments demonstrate that bubbles **can** transport particles from **the sediment** to the lake surface. "*

Line 307: Lines 134-135 mention 'On 26 June 2018 we sampled for cyanobacteria bubble transport using similar procedures, except we used a simple inverted funnel instead of a custom bubble size sensor to intercept rising bubbles' whereas Fig. S7 caption shows "Frequency distribution numbers are approximate because the bubble size sensor is unable to measure fast bubble flux or very small bubbles" – It is important to check the compatibility between these statements, particularly for data of 26 June 2018 if used.

*The data from Fig. S8 is only from the column experiments, not collected during the 26 June 2018 sampling. Fig S7 shows the frequency of bubble diameter naturally occurring (from previous work) and triggered (this work, not for the 26 June 2018 date) only collected during the October 2017 sampling event, where cyanobacteria were not measured.*

*To clarify the difference between Fig. S7 and Fig. S8, and to further clarify the figures themselves, we have updated the figure captions to read:*

*Figure S7. Frequency of occurrence of bubble diameter during triggered (purple) and natural (gray) events **in Upper Mystic Lake**. Mean (black lines) and standard deviation (shaded regions) for each event type. Bubbles were triggered by dropping an anchor multiple times during the October 2017 sampling event, while natural bubble size distribution are based on continuous measurements from the summer of 2015 and 2016 (Delwiche and Hemond, 2017). **Frequency distribution numbers for the triggered bubbles are approximate because the bubble size sensor is unable to***

*measure the rapid bubble flux that sometimes occurred with anchor-triggered bubble events.*

*Figure S8. Frequency of bubble **diameter** observed **across multiple trials (a-k) during the** cyanobacteria experiment **in the laboratory bubble column.** Some **trials** had a bimodal **diameter** distribution. Panels f and k represent trials where air was bubbled directly **above the** sediment, **and remaining panels represent trials where air was bubbled into the sediment**. Note the different y-axis scales.*

Line 317: replace ug with g.

*This mistake was corrected.*

Lines 342-343: Besides 'a significant fraction of the arsenic input to epilimnetic waters can be attributed to inflow from the Aberjona River (Hemond, 1995)' aerial transport of dust associated arsenic/metals should be invoked here to be among the unknown inputs.

*We have not properly accounted for all other forms of input of metals to the lake by restricting the next sentence to just surface water input, so we have adjusted that to include atmospheric deposition with "However, bubble-mediated fluxes of arsenic or other sediment-borne metals may represent a larger fraction of epilimnetic input in other lakes having lower influx rates from surface water inflow **or other external sources, such as atmospheric deposition (Csavina et al., 2012)**."*

Lines 361-362: 'Bubble-transported particulate matter contained cells at a rate of approximately 30 cells mL-1 gas, indicating that bubbles are capable of transporting cyanobacteria through' – May be revised as "Bubble-transported particulate matter contained cells at approximately 30 cells mL-1 gas, indicating that bubbles are capable of transporting cyanobacteria through". A 'rate' is expected to be material transferred during a specific duration (time). \*\*\*

*Thank you for identifying this error. It has been changed as suggested.*

Please also note the supplement to this comment:
https://www.biogeosciences-discuss.net/bg-2019-243/bg-2019-243-AC1-
supplement.pdf

———————————————————

[Figure]

[Figure]

*Figure S1.* Picture of the lake surface near the beginning (A) and end (B) of a triggered bubble event at 15 m depth showing an accumulation of particulate matter (visible as light specks on the water surface).

**Fig. 1.** Figure S1. Picture of the lake surface near the beginning (A) and end (B) of a triggered bubble event at 15 m depth showing an accumulation of particulate matter (visible as light specks on the water

---

## Author Comment (AC2) · 18 Feb 2020

*We would like to thank the referee for looking over this work and providing valuable critiques to our paper. The comments are thoughtful and bring up many important points, which we addressed individually below.*

Anonymous Referee 1 Authors state that the particles associated with the bubbles, almost entirely originated from the sediments, rather than from the water. Will this statement hold true in case of turbid waters? Please clarify.

*We do not actually know whether the sediment particles have been scavenged from the*

*plume of sediment in the water column, or the sediment directly. We have evidence to suggest that only a small portion ( 10%) seem to originate in the water column from the column experiments, but the concentration of particles in the water column could have been different between experimental conditions in the column and field. As such, more turbid waters could result in larger concentrations originating from the water column as compared to the sediment, but further work is needed to understand this difference.*

*We were also not clear about the water column conditions when we conducted our tests for particle scavenging in the experimental column. Because these tests were done after tests where bubbles were emitted from the sediment bed, the water column was visibly turbid and contained many suspended particles. We have added two sentences to clarify this point:*

*(In Methods)* **"Scavenging tests were conducted after particle transport tests, so the water column above the sediment bed was turbid and contained a plume of sediment particles. "**

*(In Results)* **"We conducted the scavenging tests when the water column was visibly turbid and contained a plume of suspended particles from previous tests."**

*We also add this as a possible mechanism in section 3.1:* **"These particle loadings on bubbles, and any ecosystem-wide flux estimates derived from them, must be qualified by the fact that neither triggered bubbles nor bubbles in the bubble column fully replicate natural bubbling. In particular, the triggering of bubbles with an anchor may have raised plumes of suspended sediment through which some fraction of produced bubbles had to rise, and within which the possibility of scavenging should be considered."**

Add the details of dissolved oxygen concentration, temperature and total suspended matter in the water column at the lake sampling station.

*We have added a figure (Fig. S3) showing the temperature profile taken during the*

[Figure]

*June 26, 2018 sampling event. Previous work on Upper Mystic Lake has shown that dissolved oxygen tracks closely with temperature (Delwiche and Hemond, 2017). We do not have a total suspended matter profile.*

*Delwiche, K. B., and Hemond, H. F.: Methane Bubble Size Distributions, Flux, and Dissolution in a Freshwater Lake, Environ Sci Technol, 51, 13733-13739, https://doi.org/10.1021/acs.est.7b04243, 2017.*

Did you observe any bubble breakup during the transport through the flexible tubing? If yes, does it affect the final bubble size count and volume transported?

*The bubble size sensor was placed below the sample cup set-up, which contained the flexible tubing, so any breakup within the tubing (which did occur) did not affect the measured size distribution. However, the size distribution could have been affected by rapid bubble flux, which can cause bubbles to coalesce within the funnel constriction leading to the bubble size sensor (as described in Delwiche et al, 2017). To address this fact, we have modified the text:*

*Anchor-triggered bubbles were significantly smaller (average diameter 5.6 mm) than those measured for natural bubbling events (average diameter 6.4 mm) during a 2016 field campaign [Fig. S7, (Delwiche and Hemond, 2017)], **even though relatively high bubble flux events (such as those triggered by anchor dropping) can lead to some bubble coalescence within the funnel constriction in the bubble size sensor [ (as described previously (Delwiche and Hemond, 2017)]."***

Line 114, please add the grade of $HNO_3$ used for rinsing.

*We used reagent grade $HNO_3$ for all acid washing, and have amended the text to reflect this:*

*"All sample cups were soaked in 5-10% **reagent grade** $HNO_3$ for 24 hours..."*

Authors dropped a cinderblock to trigger bubble release. Please state the difference in bubble volume during natural release and forced release.

*This information is presented in section 3.2 Triggered bubbles are smaller than natural bubbles, but both are larger than 1 mm where differences between sizes decrease, making it unlikely that their difference in size should substantially change transport.*

The impact of cinderblock on the lake floor would have re-suspended a significant amount of sediments. Does the forced release, thus suggest a much larger than natural bubble release mediated particle transport?

*We agree with the reviewer that triggering a bubble release with an anchor drop suspends a significant amount of sediments. We also wondered if this suspended sediment would artificially raise the measured rates of bubble particle transport. To address this question, we conducted the particle scavenging experiments in the bubble column, as described in section 2.3. The scavenging tests were done when the water column had significant amounts of suspended sediment from previous trials. Bubbles passing through this sediment cloud had only around 10% of the particle mass from bubbles emitted from the sediment, indicating that while particle scavenging does occur, it is relatively minor. However, we agree that anchor dropping could still influence bubble mediated particle transport, and future research is needed to assess the particle transport rates for naturally occurring bubbles.*

The collection of sediment by dredge and subsequent transport in bucket, would have resulted in the release of a significant amount of gas from the sediments. Can the authors provide the difference in the gas content of in-situ sediments and those collected by dredge and brought to the lab in a bucket?

*The gas content of the sediment was not measured, but would certainly be lower once removed from the environment by the dredge and placed into the bucket. However, the gas content of the sediment was not critical to the development of bubbles in the experimental bubble chamber. We used a syringe pump to inject gas into the sediment bed. For this reason, we did not find it critical to measure the gas content of the sediments collected in the environment.*

What was the percentage of bubbles breaking up, when striking the inverted funnel and releasing the cyanobacteria?

*As the reviewer points out, there are a number of potential experimental artifacts that could decrease the measured amount of sediment and cyanobacteria transport (including particles adhering to the sampling apparatus, as discussed earlier and now included in the manuscript). However, we have not found that bubbles break up when encountering an inverted funnel. Previous work looking at potential bubble break-up when bubbles reach the bubble sensor funnel found instead that bubble coalescence can occur when bubble flux is high enough. This coalescence relates to the reviewer's previous comment on how bubbles break up could affect size measurements, so we encourage the reviewer to see that response.*

Authors used air, instead of methane in the laboratory experiment. Will there be a difference in the particle transport by an air bubble as compared to methane bubble? Please discus in the text.

*The composition of the air in the bubble was dramatically different between the experimental column and the field, given the origins of both gases. If the experiment was conducted at high pressure, such as in the deep ocean, this difference in gas composition in the bubble could reach a critical point where it could affect the bubbles and particle transport. However, at the pressures found within our system (both lake and column), the composition of gas is unlikely to influence bubble properties or particle transport.*

*In support of the conclusion above, using either air in the column or gas from the sediment resulted in a similar amount of particle transport per ml gas ("0.01 $\pm$ 0.006 mg/mL in the bubble column, compared to 0.01 $\pm$ 0.01 mg/mL on June 2018 in the field"). However, the differences between those amounts and the amounts measured in the field in October 2017 (0.09 $\pm$ 0.07 mg/mL) are substantial, so we do not fully understand all of the factors (potentially gas composition) that influence particle trans-*

*port.*

*We also added some general caveats to this approach, which would include gas composition (e.g.):* **" There remains the possibility that our measured bubble particle transport rates differ significantly from those from naturally emitted bubbles, and this remains an important area for future research."**

**" While this variability in cell transport between column measurements and estimates of potential field transport highlights the need for continued research, it is useful to estimate the potential range of cyanobacterial transport."**

How did the authors decide the rate of injection of air into the sediments? What happened to the gases already present in the sediments when authors injected the air?

*We have added the following text to the manuscript to clarify these points:*

**"The bubbling rate was calibrated to achieve a relatively steady release of bubbles without substantial wait time in between. While we expect that much of the gas naturally existing within the sediment was released during sediment collection and as it was transferred to the sample bed (indeed we did not observe natural bubble release from the sediment bed prior to experimental trials), remaining gas could have been incorporated in to rising bubbles."**

Line 266, authors did not estimate the gas reserve in the sediments. How can they infer that the lower gas volume did not indicate a smaller gas reserve?

*As you point out, we did not measure the gas reserve in the sediment, so we cannot speculate as to the cause of the lower gas volume in June 2018. We have re-framed the section to focus on the observations and avoid undue speculation: "Both field and bubble column experiments demonstrate that bubbles* **can** *transport particles from* **the sediment** *to the lake surface.* **A** *positive correlation (p< 0.05 level for October 2017 (r2 = 0.76), p=0.15 (r2=0.38) for June 2018 ) was found between total particle mass and gas volume in bubble traps for both field sampling campaigns (Fig. 1). The gen-*

*eral magnitudes of particle loadings on bubbles in column experiments and on bubbles observed in triggered experiments in the field were of similar magnitude; 0.01 ± 0.006 mg mL -1 in the column vs 0.09 ± 0.07 mg mL-1 on October 2017 and 0.01 ± 0.01 mg mL-1 on June 2018 in the field."*

If the positing of boat influenced the bubble release, then how can they quantify the bubble volume and associated particle transport?

*It was indeed a challenge to position the boat above the sample plume, particularly when winds blew us off course between anchor drop and bubbles reaching the surface. However, since we were interested in particle transport per gas volume, our results should not be affected by whether we captured all gas from a particular bubbling event.*

*We note that this sentence is now re-written in response to other comments, as mentioned above.*

Line 273, I do not agree with the comparison of experimental column release with that from the natural lake environment. As stated above the conditions in the lab were completely different than that in the lake, and thus any comparison between the two is superfluous.

*As any controlled environment will have many differences from the natural environment, we hope that you will agree that the experimental columns were within the range observed in the field, thus can be used to verify that cyanobacteria can move quickly on these bubbles. The bubble column work was necessary to test the importance of particle shedding and scavenging (something we could not test in the field), and the fact that bubble column particle transport was of similar magnitude to field results ( 0.01 ± 0.006 mg/mL in the bubble column versus 0.09 ± 0.07 mg/mL and 0.01 ± 0.01 mg/mL in the field) indicated that the bubble column results could inform field processes. However, to acknowledge the necessary differences between the controlled and natural environments, we have added the following text:*

*"Although this is significantly higher than the measurements made in the bubble column, the conditions in the column are substantially different from the conditions in the field and the sediments used in column had been stored for 8 months, so the cyanobacteria cell concentration was 10 times less than fresh sediments. While this variability in cell transport between column measurements and estimates of potential field transport highlights the need for continued research, it is useful to estimate the potential range of cyanobacterial transport."*

Authors state a large difference in the size of natural and forced release of bubbles. Then what is the reliability of the volume and particle transport estimated by the authors?

*There is a large amount of uncertainty in amount of particle mass transported per ml of bubble volume in our measurements, which was not properly emphasized before in the manuscript. The differences in bubble size could be one aspect of this uncertainty. In response to this comment and other referee comments, we have emphasized the uncertainty in the text and removed amounts of cells or arsenic transported from the abstract. Even with these large uncertainties, we can still put our results into context by saying that we expect that this type of transport might be small compared to other inputs for arsenic, but that bubble-mediated cell transport could be a substantial part of the life cycle of cyanobacteria in this lake. This provides contexts for what should be pursued in future experiments while still emphasizing the uncertainty in our measurements. We hope that this provides better insight into the reliability of these measurements.*

Line 25, change 'Concentrations' to 'Concentration' Line 27, change 'concentrations' to 'concentration'

*We have also changed the "**A c**oncentration of 105 cyanobacteria cells mL-1 **is** considered to present a risk of both acute and chronic health effects (Backer, 2002), and many states, including Massachusetts, issue public health warnings **for recreational water bodies** when **the** cyanobacteria cell concentration **exceeds this** value."*

Line 40, modify 'et. al.' with 'et. al.'

*According to other referee comments, we have changed this sentence to "**Previous research** showed that recruitment..."*

Line 48, insert space after 2008;

*It seems that many of the references required spaces to separate them. This has been addressed here and in many other instances in the text.*

Line 71, change 'volumes' to 'volume'

*This has been changed.*

Line 74, change 'greatest' to 'a considerable'

*We agree that removing greatest is advisable, but tried to improve the sentence structure with the following "This potential transport pathway could be relative**ly more important** for metal and cyanobacteria transport in eutrophic, deep, stratified lakes, such as UML."*

Line 79, change 'distribution' to 'distribution'

*This "s" has been removed from "distribution".*

Line 119, change 'mixing from of the' to 'mixing from the'

*This has been changed to "preventing mixing **of** sediment to the surface"*

Line 123, change 'an' to 'a'

*This has been changed.*

Line 148, change 'column is comprised' to 'column comprised'

*This has been changed to "The column is composed of four section..."*

Line 176, change 'um' to $'_m$

*This has been changed.*

Line 180, change 'metals analysis on bulk sediment' to 'metal analysis in bulk sediment'

*This has been changed.*

Line 185, change 'which use' to 'with use'

*This part of the sentence has been removed.*

Line 186, change 'analysis on' to 'analysis of'

*This was changed.*

Line 188, 5 umol filter? Is it correct?

*umol was not correct and we changed to 5 um.*

Please also note the supplement to this comment:
https://www.biogeosciences-discuss.net/bg-2019-243/bg-2019-243-AC2-supplement.pdf

[Figure]

[Figure]

**Fig. 1.** Figure S3. Water temperature profile taken during June 16, 2018 sampling event on Upper Mystic Lake.

---

## Author Comment (AC3) · 18 Feb 2020

*We would like to thank the referee for looking over this work and providing valuable critiques to our paper. The comments are thoughtful and bring up many important points, which we addressed individually below.*

Anonymous Referee 4

The issues with sample collection make me call into question the quantitative results and budget. Please see my specific comments below for further details. Ultimately, the data need to be published, but the manuscript needs major revisions to remove

the budgets which are likely inaccurate, given the sample collection procedure. Please refocus the manuscript to state the observations and cast your results in light of how the samples were collected.

*We agree that the quantitative results and budget analysis are highly speculative, so the suggestion of removing the budget analysis would certainly be one way of addressing this issue. However, we propose keeping the budget calculations in the text, but making sure to emphasize the proper uncertainty associated with these budget estimates and to replace any specific estimates highlighted in the abstract or conclusions with a statement that more work is needed to calculate a proper budget for this mechanism. We hope that this approach would provide some context for the observations while remaining realistic about the fact that the information isn't at the level it needs to be for estimating a proper budget. We hope that our revisions have captured the spirit of this comment, while still providing some context to interpret our observations and to inspire future research.*

Specific Comments: L 23-24: Define "problematic". What does this mean for cyanobacteria? Be more specific.

*This statement was clarified as "In a 2012 national assessment, 15.2% of surveyed lakes in the U.S. were **categorized as Most Disturbed due to the concentration of cyanobacteria, a significant increase in lakes with this categorization (8.3%, 95% confidence intervals 4.0-12.5%)** over the 2007 assessment (U.S. Environmental Protection Agency, 2016)."*

L 29-30: What about the "improved understanding"? What type of understanding? Be specific.

*We have changed this to be more specific as "**Identifying** the sources and mechanisms of transport of these substances within lake ecosystems can help predict the fate of contaminants and aid remediation efforts."*

L 110-111: How do you know the bubble transported biology and chemistry is no adhered to the inner walls of sampling equipment? Do your measurements represent an underestimate?

*This is a point that was also brought up by a previous referee, so we have added a comment about this potential sampling artifact, which would underestimate transport:*

**"The interaction of bubbles with the flexible tubing resulted in visible particle attachment to the tubing, making our estimates of particle mass transport a lower bound."**

L 172-173: Are these filter measurements meant to be volumetric? If so, do you know how much water passed through each filter before clogging?

*For these filter measurements, we recorded the total volume filtered and the total mass accumulated, whether or not this was distributed over more than one filter because of clogging. Thus, we do not know the volumes passed through individual filters, only the total volume of water associated with a total particle mass.*

*We have amended our text to read: "Due to filter clogging, we typically used multiple filters for each **sample, and total particulate transport per sample was calculated by summing the particle mass on each filter and dividing by the total gas volume associated with the sample.** "*

L181: I don't know how this relates to the accuracy and precision of your measurements? How do counts per second relate to concentration?

*The relative standard deviation of the ICP-MS counts relates to the uncertainty in the measurements. The uncertainty for the sediment digests is quite low, and while it is higher in the less concentrated bubble transported particle samples, this uncertainty is still low relative to the experimental uncertainty. We have added the following line to the text:* **"These relatively low RSD values indicate that analytical uncertainty is low, especially compared experimental uncertainty."**

L 266: This is an excellent study and I think your experiments and testing shows bubbles play a role in lakes that has not been considering from a biological perspective. This study needs to be published, but I can't get over the anchor drop issue. I have thrown many anchors overboard in lakes and the plume of sediment is always significant. I have a hard time decoupling this disturbance with your results. There needs to be a paragraph describing how the laboratory results follow the lake results and the anchor had minimal impact on the lake results. Although, your laboratory results show sediment disturbance impact the bubble transported particles. How can you decouple these methodological problems with your results? What if you shift the focus of your manuscript to documenting that bubbles DO transport chemistry and biology, but stop short of the full budgets, as I think those are biased due to the methodological problems.

*We agree with the reviewer that triggering bubbles with an anchor drop leads to substantially different conditions than naturally ebullition. We wish we could have collected samples from natural ebullition alone, but this would have resulted in long wait times and probable changes in the cyanobacteria population prior to sample analysis. We attempted to alleviate some of this concern by using the laboratory bubble column experiments to demonstrate that particle scavenging when bubbles rise through a plume of sediment is still a relatively minor contribution to total particle transport. However, we agree that this experiment alone cannot account for all potential effects of the anchor drop. We feel this is an excellent area for future research, either in systems with much higher ebullition rates such that natural bubbles could be used, or potentially with updated experimental apparatus that can utilize natural bubbles.*

*To address these concerns, we have re-worded the text in numerous areas to highlight the uncertainty while still providing context for whether these observations could substantially impact chemical cycling or cyanobacterial life cycle. Some examples include:*

*Abstract- "* ***Although more work is needed to reduce uncertainty in budget estimates****, bubble-facilitated cyanobacterial transport* **has the potential to contribute**

*substantially to the cyanobacteria cell recruitment to the surface of* this lake and *may thus be of particular importance in large, deep, stratified lakes."*

*Results- "**These particle loadings on bubbles, and any ecosystem-wide flux estimates derived from them, must be qualified by the fact that neither triggered bubbles nor bubbles in the bubble column fully replicate natural bubbling. In particular, the triggering of bubbles with an anchor may have raised plumes of suspended sediment through which some fraction of produced bubbles had to rise, and within which the possibility of scavenging should be considered."***

*" **However, many questions remain regarding bubble-mediated transport in natural systems, including how the change in water density at the thermocline affects bubble rise and associated chemical and biological material."***

*"**There remains the possibility that our measured bubble particle transport rates differ significantly from those from naturally emitted bubbles, and this remains an important area for future research. However, despite this uncertainty, broadscale estimates of arsenic and cyanobacteria cycling can provide important context as to whether these processes may be significant in UML."***

*" **These calculations demonstrate that bubble transported cyanobacteria could negatively impact water quality, though more research is warranted to improve these estimates."***

*" Using the maximum observed recruitment rate of 2.3 x 105 cells m-2 day-1 (Brunberg and Blomqvist, 2003) from sediments for the area of the lake above 12 meters, we estimate that bubbling could contribute 14 % of cyanobacterial recruitment in the lake, **but 95% confidence intervals range from less than 0 to 46% of overall recruitment. While we cannot rule out the possibility that this is an insignificant source of cells given the large uncertainty in these measurements, the potential for bubble-mediated transport to contribute substantially to the source of cyanobacteria cells at the lake surface warrants further investigation."***

*Conclusions- "Bubble mediated transport of cyanobacteria cells may contribute **substantially** to cellular recruitment from the sediment, **but the uncertainties in our measurements make these estimates speculative."***

L 268-270: This observation is baseless since you caused the ebullition.

*The reviewer makes a good point that natural variation in ebullition has nothing to do with the variation in mass transport observed in our triggered bubbling events. We have removed this sentence.*

L 277-280: This is analogous to dropping an anchor on the lake sediments. How do you reconcile these laboratory experiments with what you did in the field? Again, this is evidence the focus of the manuscript should be focused to an observation that bubbles do transport chemistry and biology, but do not calculate budgets because the evidence shows they are not accurate.

*Two observations from the columns with recently disturbed sediments (similar to the anchor drop, as mentioned in the comment) are similar to those with "normal" sediment, so the impact of these disturbances creates a complicated relationship with particle transport that we can not fully understand. The combination of both measurements ("normal" and "recently disturbed") resulted in transport that were similar to one field collection date, so it is at least in a similar range to what is occurring in the field.*

*This comment again highlights the uncertainty in our measurements. We agree with this comment and address it by making the uncertainty in our calculations more prominent, downplaying numbers in the abstract and conclusions, but keeping the budgets for context. We have re-written the text in numerous locations to highlight sources of uncertainty (mentioned above). However, we do still see value in budget calculations, however uncertain they may be. For example, the rough budget calculations for arsenic show a several order of magnitude gap between potential bubble arsenic transport rates and other transport rates within UML, indicating that even if our estimates are biased low, they are unlikely to be high enough to matter in UML. Conversely, the*

*upper threshold for cyanobacteria transport in UML does fall within the realm of an important flux, which is a justification for further research in this area. We therefore think these estimates give a useful perspective, but we emphasize the large uncertainty that exists in these measurements and that the budgets are a best guess.*

L 283-285: Were there particles to scavenge? This was tap water, right?

*This reviewer and one other have helpfully pointed out that we were not clear about the water column conditions when we conducted our tests for particle scavenging. As discussed previously in this response, scavenging tests were done after tests where bubbles were emitted from the sediment bed, so the water column was visibly turbid and contained many suspended particles.*

*We have added a sentence to clarify this point:* ***"We conducted the scavenging tests when the water column was visibly turbid and contained a plume of suspended particles."***

Section 3.3 header: Again, I have a hard time reconcile the topic of this section that particles originated in the sediment after traveling through a plume of sediment. Maybe scavenging is a more active process and makes up a larger percentage of the particles when not passed through a plume of sediment.

*We agree that bubble scavenging of particles within the water column could contribute to the particle burden, and thus not all particles originate in the sediment. Indeed, our scavenging tests shows that approximately 10% of the particles transported to the surface could be picked up within the relatively turbid water column. This indicates that within our experiments, a substantial fraction of the particles appear to come from the sediment bed itself. However, as pointed out previously, the artificial conditions for bubble release in both our laboratory and field experiment could influence our results. To acknowledge this uncertainty, we have changed the section title to:*

*"3.3* ***Bubble-transported particles have chemical and biological characteristics***

***similar to sediment*** *The data on bubble particle mass transport clearly shows that bubbles are capable of transporting particles from relatively deep depths, and minimal rates of particle shedding and scavenging in the water column* **suggests** *that these particles originate* **primarily** *in the sediment. "*

L 325-326: Observations like this are the reason this manuscript needs to be published.

*We appreciate your support for the publication of this work. To highlight the finding of potential ephippia in the particles, we have added a reference to the specific panel in Figure S10 that may show ephippia (Fig. S10-B).*

L 353-354: This is a major finding of this study and should be a highlight.

*We appreciate the reviewer's enthusiasm for the content. The referenced sentence in L 353-354 speculates that since cyanobacteria overwinter in the lake sediments, bubble-mediated transport could be a mechanism of inoculating the upper water column with these cells. We believe we have highlighted this possibility with the mass budget calculations that compare potential bubble cell transport to other methods of cell recruitment. However, as discussed previously in responses to this reviewer, there remains a high degree of uncertainty around our estimated cell flux.*

L 374: What does it mean to have a negative rate of transport? Are bubbles actually sequestering cells from the surface waters? This is another reason why I think the budgets need to be removed and the focus placed on the observations and laboratory experiments.

*A negative transport rate is not meaningful, but is another aspect of the variability of our measurements that add uncertainty to the budgets. As discussed earlier, we agree with the reviewer that more attention should be given to the uncertain nature of our budget calculations, and have re-written portions of our text accordingly. Furthermore, we now conclude with the statement that: "Using the maximum observed recruitment rate of 2.3 x 105 cells m-2 day-1 (Brunberg and Blomqvist, 2003) from sediments for*

*the area of the lake above 12 meters, we estimate that bubbling could contribute 14 % of cyanobacterial recruitment in the lake, **but 95% confidence intervals range from less than 0 to 46% of overall recruitment. While we cannot rule out the possibility that this is an insignificant source of cells given the large uncertainty in these measurements, the potential for bubble-mediated transport to contribute substantially to the source of cyanobacteria cells at the lake surface warrants further investigation."***

L 400: Given the large errors in your bubble transport of cells, I have a hard time following how the error now is so small. The error propagation is not well explained.

*This is an error, and the range of values reported comes from using both 9 meters and 12 meters as the cut-off for where cyanobacteria would be able to recruit to the surface without bubbles. We agree that this does suggest a smaller uncertainty in the final budget than is warranted from the data.*

*To address this and the comment from above, we propose to still include the budgets in the presentation of the data for perspective, but to better emphasize the speculative nature of these budget results and the uncertainty associated with it. This provides context for the results and motivates additional research in the future on this topic, while still being realistic about whether these transport rates are well constrained. Even with the large uncertainty in particle transport values, the arsenic transport is unlikely to be a substantial part of arsenic found in the lake surface, but bubbles could still be an important part of cyanobacteria transport.*

*Since there are a number of uncertainties associated with cyanobacteria transport, we can emphasize that bubble-mediated transport has the potential to be a significant source of cell recruitment, especially in deep, eutrophic lakes. However, more work is needed to better constrain these values to determine the actual contribution.*

Technical Comments: L 22: Delete "are". *Thank you for finding this glaring error in our first sentence, we have deleted the "are".*

L 22-23: First sentence needs a citation.

*We have added two references that provide an overview of how water quality is a wide-spread phenomenon that will be likely exacerbated with increases in urbanization and climate change. "Deterioration of water quality is wide-spread and expected to become more acute with increased urbanization and climate-change* ***(Zhang, 2016; Paerl et al., 2011)."***

L 32-34: First sentence of the paragraph, poor sentence structure, please rewrite.

*We have clarified this sentence to read:*

*"Because sediments are typically major repositories of contaminants (Nriagu et al., 1996; Pan and Wang, 2012; Taylor and Owens, 2009),***it is important to understand the processes leading to contaminant mobilization."***

L 35-37: "However, transport to surface: : :" Poor sentence structure, please rewrite.

*We agree that this sentence was poorly worded. We have restructured the whole paragraph to improve readability:*

*"Because sediments are typically major repositories of contaminants (Nriagu et al., 1996; Pan and Wang, 2012; Taylor and Owens, 2009),* ***it is important to understand the processes leading to sediment mobilization****. Metals can be mobilized from sediments via solubilization by oxidation-reduction reactions, and by sediment resuspension,* ***acidification*** *or bioturbation (Calmano et al., 1993; Eggleton and Thomas, 2004; Schaller, 2014; Schindler et al., 1980).* ***Likewise, over-wintering cyanobacteria and algae concentrated in the sediments are mobilized through germination, wind-induced resuspension, or bioturbation (Ramm et al., 2017; Verspagen et al., 2004; Stahl-Delbanco and Hansson, 2002)****. In some cases, the number of resting cells in sediment can be predictive of the severity of subsequent bloom events (Anderson et al., 2005).* ***Previous research*** *showed that recruitment from sediments of the potentially toxic cyanobacterium Microcystis was a major driver of the summer bloom*

*(Verspagen et al., 2005). Cyanobacterial recruitment to surface waters from deep sediments is expected to be inhibited by stratification, low oxygen concentration, and low light levels (Ramm et al., 2017).* **Metals mobilized from sediment under stratified water columns will also be inhibited from reaching surface waters due to stratification (Wetzel, 2001)."**

45-46: "Bubbling from anoxic sediment: : :" Sentence missing numerous citations.

*Thank you for bringing this to our attention, we have added the following two citations showing substantial contribution of methane bubbling to total freshwater emissions: Bastviken, D.; Tranvik, L. J.; Downing, J. A.; Crill, P. M.; EnrichPrast, A. Enrich- prast, A. Freshwater methane emissions offset the continental carbon sink. Science 2011, 331, 50−50.*

*Deemer, B.; Harrison, J.;Li, S.; Beaulieu, J.;DelSontro, T.; Barros, N.; Bezerra-Neto, J.; Powers, S.; Dos Santos, M.; Vonk, J. Greenhouse gas emissions from reservoir water surfaces: A new global synthesis. BioScience 2016, 66 (11), 949−964.*

*Citations for the ability of bubbles to transport particles are already provided in subsequent sentences detailing this process in industry and marine systems.*

L 50- 53: "Bubble-mediated particle: : :" Poor sentence structure, confusing, please rewrite.

*We agree this sentence was quite poorly written, and have changed it to:*

**"Bubble-mediated particle transport also occurs in the open ocean where bubbles are injected into the water by breaking waves, scavenge surface-active particles as they rise, and then deposit these particles on the ocean surface (Aller et al., 2005; Blanchard, 1975; Wallace et al., 1972; Liss, 1975)."**

L184-185: "We filtered bubble: : :" I did not understand this sentence.

*We agree this sentence is confusing, and have shortened it to say:*

*"We filtered bubble column samples using pre-weighed 5.0 μm and 0.2 μm Whatman Nuclepore membrane filters (47mm diameter)."*

L 187: How much lower are the blanks? Actual numbers would be better. Two orders of magnitude can range from 110-fold lower to 900-fold lower. These are very different blanks.

*To clarify the blank question, we have calculated that the Whatman filters contained less than a nanogram of arsenic contamination, far below the sample concentrations. For the Nucleopore membranes, the 5 μm filters had arsenic levels below the ICP-MS detection limit, and the 0.2 μm filters had 0.003 ± 0.002 μg per filter for the 0.2 μm filter (less than 1% of the arsenic found in the least concentrated sample). We have added the following text:*

**"Duplicate analysis of clean Nuclepore membranes (blank) was used to determine arsenic contamination of the filters and was below the detection limit for the 5 μm filters and 0.003 ± 0.002 μg per filter for the 0.2 μm filter (less than 1% of the arsenic found in the least concentrated sample). "**

L 249: mL-1 gas volume or mL gas volume-1?

was changed to "mL gas volume-1"

L 250: Estimate – estimated (past tense).

*Thank you, we have made this change.*

L 258: Bring eq. 1 up so that the reader knows the equation before getting the variables.

*We have Equation 1 to the top of the paragraph, along with a summary description of each variable to aid in readability.*

Rewrite the part about the depth interval for germination. I was lost.

*We have improved the readability of this section as: "**We conservatively assumed that germination could occur to a depth of 12 meters based on typical** light, temperature, and oxygen **levels observed in** UML (Varadharajan, 2009). The fraction (Fg) of the surface area (SA = 580,000 m2) of lake above 12 meters that could support cyanobacterial recruitment through germination is**approximately** 0.50 (Varadharajan, 2009)."*

L 362: This is a concentration, not a rate.

*Thank you, we have eliminated "a rate of".*

L 365: Keep units consistent. Use slash or exponent throughout.

*Thank you for noticing this inconsistency. We have used exponents throughout.*

Please also note the supplement to this comment:
https://www.biogeosciences-discuss.net/bg-2019-243/bg-2019-243-AC3-supplement.pdf

———————————————

---

## Author Response (AR2)

Comments to the Author:
One of the referees' has made some useful suggestion. Also, there are minor editorial comments. Authors may like to incorporate these suggestions in the manuscript.

Editorial comments:
Figure 5: The figure caption is somewhat confusing to a general reader. Lines 638-639 should read as "The background concentration of cyanobacteria cells in the water column was initially low -----. What is concept of sample type given as "Just air"? Authors may like to use more appropriate scientific/technical terminology to describe the experimental set-up.

*Thank you for identifying the grammatical mistake on lines 638-639- they have been changed. We agree that the description of sample types in the graph was not an appropriate scientific description of the samples. We changed it to "Bubbling above sediment" and "Bubbling within sediment" to describe where bubbles originated from in the experiment and changed the figure caption as:*

**Figure 5:** *The concentration of cyanobacteria cells (as measured by quantitative PCR) increases in the experimental water column and bubble traps after initiating bubbling within sediments. The background concentration of cyanobacteria cells in the water column was initially low ("Before bubbling") but increased after bubbling air through the sediment. The concentration of cells in the bubble trap increased even if bubbles do not pass directly through sediment, but instead originate above the sediment bed ("Bubbling above sediment"), from cells contaminating the surrounding water column. However, the highest concentration of cyanobacteria in the bubble trap was observed when initiating bubbling from within the sediment ("Bubbling within sediment") from direct transport of cells from the sediment into the bubble trap. The increase in cell concentration in both the water column and the bubble trap after bubbling within sediment is evidence for cyanobacteria transport via bubble floatation. Error bars show standard deviation across measurements.*

The authors have improved their manuscript. I had asked for greater specificity and clarification in their writing and that was completed. I also asked the authors to provide qualifying statements to frame their results and conclusions. Now the manuscript makes the appropriate conclusions given their methods and results. The paper forms a compelling hypothesis for further testing, making these data valuable. Thanks to the authors for the improvements to their manuscript and providing detailed responses to my concerns.

Minor corrections
L365-367: Not sure if the doom and gloom statement of swimmers ingesting Arsenic is warranted at this time. This is one of those conclusions that could be taken out of context in the wrong way. Your choice, but I suggest removing. The solid conclusion based on the data occurs at the end of that section.

*Given the modest importance of bubbling in the arsenic cycling in this lake, this statement is a valid concern, and we have removed it from the manuscript.*

Figure 4 (a) and (c): Scale on Y-axis should start from 0.0 instead of – 0.2 and – 0.5. What is the concept/significance of negative values?

*The scale of the Y-axis was changed to start at 0. No negative values were observed.*

[revised manuscript text omitted]

Microsoft Office User 4/22/20 9:29 AM

Unknown

Microsoft Office User 4/21/20 4:26 PM

Microsoft Office User 4/21/20 4:26 PM

Microsoft Office User 4/21/20 4:37 PM

Microsoft Office User 4/21/20 4:37 PM

Microsoft Office User 4/21/20 4:37 PM

Microsoft Office User 4/21/20 4:33 PM

Microsoft Office User 4/21/20 4:33 PM

**Figure 5:** The concentration of cyanobacteria cells (as measured by quantitative PCR) increases in the experimental water column and bubble traps after initiating bubbling within sediments. The background concentration of cyanobacteria cells in the water column was initially low ("Before bubbling") but increased after bubbling air through the sediment. The concentration of cells in the bubble trap increased even if bubbles do not pass directly through sediment, but instead originate above the sediment bed ("Bubbling above sediment"), from cells contaminating the surrounding water column.

However, the highest concentration of cyanobacteria in the bubble trap was observed when initiating bubbling from within the sediment ("Bubbling within sediment") from direct transport of cells from the sediment into the bubble trap. The increase in cell concentration in both the water column and the bubble trap after bubbling within sediment is evidence for cyanobacteria transport via bubble floatation. Error bars show standard deviation across measurements.